# Ultra Lowrate Image Compression with Semantic Residual Coding and Compression-aware Diffusion

Anle Ke [1]  Xu Zhang [1]  Tong Chen [1]  Ming Lu [1]  Chao Zhou [2]  Jiawen Gu [2]  Zhan Ma [1]

## Abstract

Existing multimodal large model-based image compression frameworks often rely on a fragmented integration of semantic retrieval, latent compression, and generative models, resulting in suboptimal performance in both reconstruction fidelity and coding efficiency. To address these challenges, we propose a residual-guided ultra lowrate image compression named **ResULIC**, which incorporates residual signals into both semantic retrieval and the diffusion-based generation process. Specifically, we introduce **Semantic Residual Coding (SRC)** to capture the semantic disparity between the original image and its compressed latent representation. A perceptual fidelity optimizer is further applied for superior reconstruction quality. Additionally, we present the **Compression-aware Diffusion Model (CDM)**, which establishes an optimal alignment between bitrates and diffusion time steps, improving compression-reconstruction synergy. Extensive experiments demonstrate the effectiveness of ResULIC, achieving superior objective and subjective performance compared to state-of-the-art diffusion-based methods with -80.7%, -66.3% BD-rate saving in terms of LPIPS and FID. Project page is available at https://njuvision.github.io/ResULIC/.

## 1. Introduction

In recent years, learning-based image compression techniques (Ballé et al., 2018; Minnen et al., 2018; Chen et al., 2021; Lu et al., 2022; Duan et al., 2023) have gained considerable attention and shown superior performance compared to traditional codecs, such as JPEG2000 (Taubman et al., 2002) and VVC Intra Profile (ITU-T & ISO/IEC, 2020),

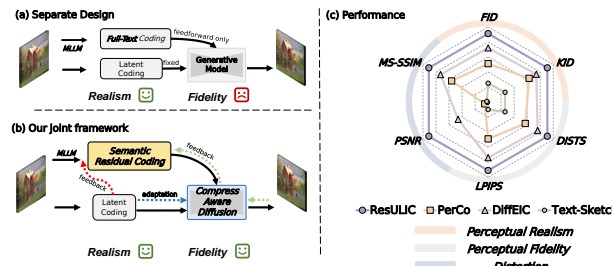

*Figure 1.* (a) The separate design for existing frameworks. The MLLM indicates the Multimodal Large Vision-Language Model. The latent coding represents the compression of features within the latent space. (b) Our pipeline with proposed Semantic Residual Coding and Compression-aware Diffusion Model. (c) Comparison with existing diffusion-based ultra lowrate image compression methods on CLIC2020 dataset.

in both objective metrics and subjective evaluations. However, at lower bitrates, these methods often struggle with excessively smooth textures or the loss of fine details and structural information.

In response to these challenges, extensive research has been conducted on optimizing perceptual quality. Among them, Generative Adversarial Network (Goodfellow et al., 2014) (GAN)-based approaches (Mentzer et al., 2020; Muckley et al., 2023) demonstrated competitive performance with visually-pleasing reconstruction. Some works have further attempted to enhance the visual reconstruction quality by incorporating text guidance. Within the conventional encoder-decoder framework, they have integrated semantic information into either the encoding process (Lee et al., 2024) or both the encoding and decoding processes (Jiang et al., 2023) to improve the perceptual quality of reconstruction.

More recently, the advent of diffusion models (Ho et al., 2020; Rombach et al., 2022) has provided a turning point for this predicament. Existing diffusion model-based image compression methods (Careil et al., 2024; Lei et al., 2023a; Li et al., 2024a) have shown more impressive results than GAN-based methods, achieving high visual quality reconstruction at extremely low bitrates ($< 0.01$ bits per pixel (bpp)). However, the reconstruction reliability (indicating consistency and fidelity) remains unsatisfactory, with significant differences from the original inputs.

[1]School of Electronic Science and Engineering, Nanjing University [2]Kuaishou Technology. Correspondence to: Tong Chen <chentong@nju.edu.cn>.

*Proceedings of the 42ⁿᵈ International Conference on Machine Learning*, Vancouver, Canada. PMLR 267, 2025. Copyright 2025 by the author(s).

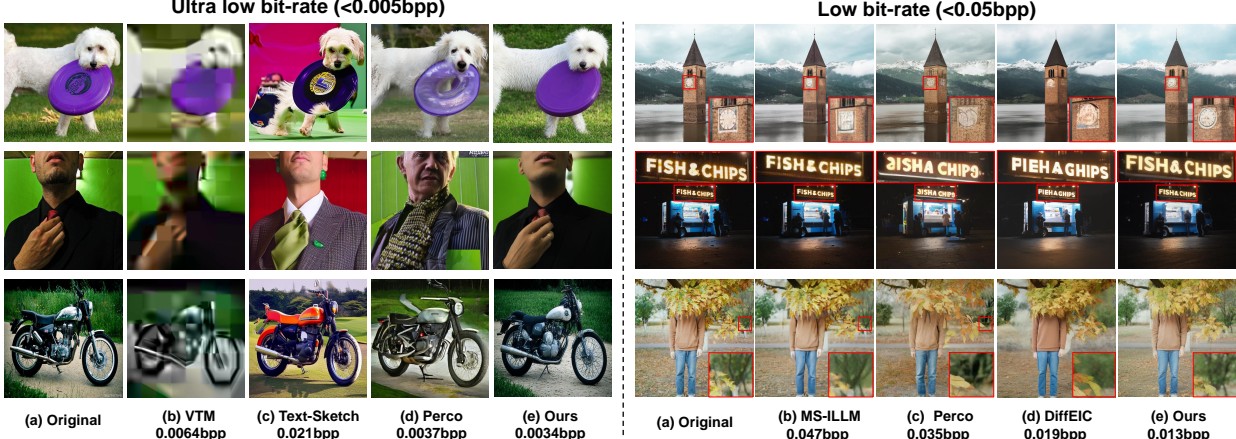

*Figure 2.* Visual comparison at extremely low bitrates. The bitrate is averaged over samples for each tested method. The major structure on the left ($< 0.005$ bpp) and the details of the text, hand, and clock on the right ($< 0.05$ bpp) are better preserved.

From a compression standpoint, the objective is to extract compact and accurate information with minimal bitrate consumption, while ensuring the reconstructed images with balanced perceptual realism and fidelity. However, two key challenges hinder the effective integration of generative models into image compression tasks:

- ***Multimodal Semantic Integration****: The effective integration of multimodal semantic information, minimizing redundancy while ensuring high perceptual fidelity at extremely low bitrates.*

- ***Compression-Generation Alignment****: Modeling the compression ratio with the noise scale in the diffusion process, enabling efficient and consistent reconstructions across varying compression levels.*

For ***challenge 1***, as shown in Figure 1(a), existing Multimodal Large Language Models (MLLM)-based methods primarily focus on simply integrating information from both texts and other content (such as sketches, color maps, or structures) to reconstruct images, overlooking the semantic information already embedded in both sources, leading to semantic redundancy. To address this problem, we propose to implement a semantic residual coding module as in Figure 1(b) into our multimodal image compression framework, aiming to achieve overall minimal bitrate consumption. Besides, to optimize the perceptual fidelity, we propose a differential prompt optimization strategy to find the optimal text prompts for improving the reconstruction consistency.

For ***challenge 2***, the degradation introduced by compression and the diffusion noising process share a common characteristic: as noise increases (or the compression ratio becomes higher), less information is preserved in the degraded image. Consequently, the compression ratio aligns inherently with the diffusion time steps. In this context, we aim to model this correlation. As illustrated in Figure 1(b), we incorporate the latent residual into the diffusion process and propose a

Compression-aware Diffusion Process, which effectively enhances reconstruction fidelity while significantly improving decoding efficiency.

Experimental results show that our model achieves both objectively and subjectively pleasing results compared to current state-of-the-art approaches, with significantly reduced decoding latency. The contributions of our work can be summarized as follows:

- We propose the Semantic Residual Coding (SRC), implementing multimodal large models as the residual extractor to remove the redundant semantic information between the original image and the compressed latent features, achieving joint bitrate reduction with efficient token index coding. A differential prompt optimization method can be further applied to efficiently search for text prompts with improved fidelity.

- We propose the Compression-aware Diffusion Model (CDM), which modeling the relationship between diffusion time steps and the bitrate of compressed images, significantly enhancing the reconstruction fidelity while reducing decoding latency.

- Based on the above key modules, a high-fidelity Residual-guided Ultra Lowrate Image Compression framework named *ResULIC* is proposed, achieving impressive visual quality at ultra-low bitrates, outperforming existing SOTA method PerCo by -80.7% and -66.3% BD-rate saving in terms of LPIPS and FID.

## 2. Background

Recent advancements in learned image compression, such as Ballé et al. (2017) and subsequent works (Ballé et al., 2018; Minnen et al., 2018; Chen et al., 2021; He et al., 2022; Lu et al., 2022; Duan et al., 2023), have showcased the potential of neural networks and advanced features like hyperpriors

*Figure 3.* **ResULIC Overview**: (1) The feature compressor transforms the original image $x$ into the compressed latent feature $z_c$. (2) The **Semantic residual retrieval (Srr)** generates optimized captions by analyzing both the decoded image $x'$ and the original $x$, with the plugin play module **Perceptual fidelity optimizer (Pfo)** to further improve reconstruction quality. (3) Text tokens are embedded into $c$ and combined with $z_c$ as conditions for the **Compression-aware Diffusion Model (CDM)** to generate the final image $x_r$.

and context models to significantly improve rate-distortion performance. Meanwhile, GAN-based approaches (Mentzer et al., 2020; Pan et al., 2022; Muckley et al., 2023) have focused on optimizing the reconstruction fidelity, especially for low-bitrate scenarios. Most Recently, compression algorithms leveraging large-scale pre-trained generative models, such as Diffusion, have emerged, demonstrating highly competitive performance.

**Diffusion-based Generative Models.** Diffusion models, inspired by non-equilibrium statistical physics (Sohl-Dickstein et al., 2015), have achieved great success in visual tasks through advancements like DDPM (Ho et al., 2020) and LDM (Rombach et al., 2022), while strategies such as adding trainable networks (Zhang et al., 2023; Mou et al., 2023; Zavadski et al., 2024) enable efficient fine-tuning of pretrained models to reduce resource requirements.

While diffusion models and their fine-tuning strategies have shown significant promise, an additional challenge lies in ensuring the reliability of generated content, particularly for tasks like compression. Previous works, such as textual inversion (Gal et al., 2023) and Dreambooth (Ruiz et al., 2023), use *soft prompts* (continuous, learnable vectors optimized for specific tasks) optimized for visual similarity, but these are not suitable for compression due to their high-dimensional nature. Other approaches, like PEZ (Wen et al., 2023), utilize discrete prompt optimization by projecting learnable continuous embeddings into the space of discrete embedding vectors to perform prompt optimization. The final optimized *hard prompts* remain in textual form and are capable of producing highly consistent images.

In recent years, there has been a notable increase in the use of diffusion models for image restoration. Prominent works such as ResShift (Yue et al., 2024) and RDDM (Liu et al., 2024b) enhance the diffusion process by incorporating the residuals between the original and degraded data. This approach significantly enhances reconstruction quality. Additionally, some works like PASD (Yang et al., 2024), StableSR (Wang et al., 2024a), start the diffusion process from low-quality images instead of pure noise to enhance efficiency and accelerate sampling. PASD relies solely on

low-quality initialization during inference, resulting in a decoupled and suboptimal process. Methods like ResShift and RDDM introduce custom noise schedulers, which require full retraining and are incompatible with off-the-shelf diffusion models. Taking a step further, we propose a bitrate-aware residual diffusion scheme specifically designed for image compression, retaining the original noise scheduler for compatibility. Additionally, our evaluation demonstrates that dynamically adjusting diffusion time steps based on compression ratios enables a more efficient framework.

**Diffusion-based Compressors.** Several diffusion-based image codecs have been proposed recently (Theis et al., 2022; Ghouse et al., 2023; Hoogeboom et al., 2023; Yang & Mandt, 2023; Ma et al., 2024; Xu et al., 2024). These methods utilize diffusion models to achieve good performance at relatively high bitrates (bpp > 0.1). With the emergence of latent LDMs (Rombach et al., 2022), Lei et al. (2023b); Li et al. (2024a;b) employed it for low-bitrate compression, further enhancing reconstruction quality. Similarly, Careil et al. (2024) integrated vector-quantized image features and captions generated by a feed-forward model (Li et al., 2022) to improve compression performance. Overall, current diffusion-based codecs have begun to show outstanding performance, but the fidelity gap between AI-generated and original content persists, and the coding efficiency of these frameworks has not been fully explored at low-bitrate scenarios.

## 3. Framework Overview

The framework is illustrated in Figure 3. It can be divided into three major parts: the feature compressor, the semantic residual coding (Sec. 4) and the bitrate-aware diffusion model (Sec. 5).

The overall compression target consists of two interrelated parts: the *latent features*, which contain information such as structures, textures, and contours, and the extracted *texts*, which capture the remaining semantic information in the image. Two compressors are proposed to handle these components in a coordinated framework. The Feature Compressor first map the image $x$ to latent space, obtaining the latent

feature $z_0 = \mathcal{E}(x)$. A neural compression network with similar architecture like (He et al., 2022) with encoder $E_\phi$ and decoder $D_\theta$ is then implemented to compress the latent features.

$$z_c = C_\Theta(z_0) = D_\theta(E_\phi(z_0)) \tag{1}$$

The reconstructed $z_c$ will serve as a guiding condition for Visual Adapter to control the diffusion to produce the final high-quality reconstruction. Detailed description can be found in the Appendix C.1.

$$x' = \mathcal{D}(z_c), \quad c = \mathbf{f}_{mllm}(x, x'),$$
$$R = R_c + R_{z_c}, \tag{2}$$
$$x_r = \mathcal{D}(\mathbf{f}_{dm}(c, z_c, N)).$$

Here $R$ represents the overall bitrates combining texts and latents. The texts $c$ is retrieved through MLLM model $\mathbf{f}_{mllm}$. $x'$ represents the intermediate reconstruction by the latent decoder $\mathcal{D}$. Final reconstructed image $x_r$ is obtained by a diffusion model $\mathbf{f}_{dm}$ and the latent decoder.

## 4. Semantic Residual Coding (SRC)

As aforementioned, existing methods usually rely on MLLM to extract full-text descriptions of the original image as conditional guidance for compression or post-processing. Take postprocessing-based frameworks as an example: no bits are needed to transmit the text, as it is reconstructed on the decoder side. However, lost semantics cannot be accurately restored.

$$c = \mathbf{f}_{mllm}(x'), \quad R_c = 0. \tag{3}$$

Meanwhile, many existing attempts based on diffusion separately transmit two bitstreams for image and text, the retrieved texts can be redundant since much of the information is already contained in $x'$, resulting in unnecessary bitrate overhead.

$$c = \mathbf{f}_{mllm}(x), \quad R_c \neq 0. \tag{4}$$

It would become an obvious drawback for compression tasks especially when the total bitrate is low. Two modules are proposed to address this issue.

### 4.1. Semantic residual retrieval (Srr)

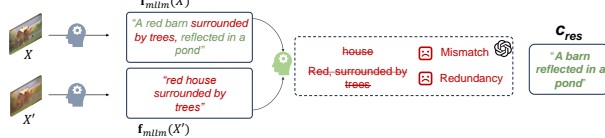

*Figure 4.* The process of **Semantic Residual Retrieval (Srr)** involves implementing the MLLM to remove redundant text and correct inconsistencies.

In Figure 4, we redesign the caption retrieval pipeline. In addition to using the original image to obtain full caption as

common practice, we also get the caption from the decoded image directly from the decoded compressed latent feature. Subsequently, we input both captions into an LLM, which outputs semantic information present in the original image but missing in the compressed image as

$$c_{\mathbf{res}} = \mathbf{f}_{mllm}(x) \ominus_{mllm} \mathbf{f}_{mllm}(x'), \tag{5}$$

where $\ominus_{mllm}$ indicates the residual retrieval process realized by LLM. This process enables the extraction of precise and compact textual conditions, allowing for adaptive optimization of bitrate allocation between text and latent representations. A detailed demonstration is provided in Section 6.2 in Figure 10. As the latent bitrate $R_{z_c}$ increases, $x' \to x$ and $R_c \to 0$, our framework approaches pure postprocessing. Conversely, as $R_{z_c}$ decreases, $x' \to 0$ and $R_{c_{res}} \to R_c$, requiring more information to be encoded within the semantic representation.

### 4.2. Perceptual fidelity optimization (Pfo)

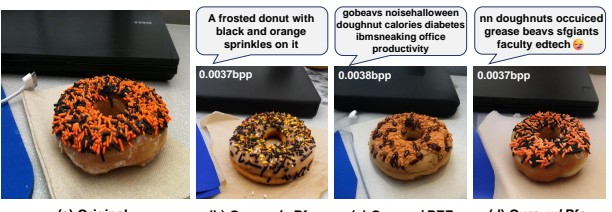

*Figure 5.* Demonstration of **Pfo** optimized prompts and corresponding reconstruction. The prompts after optimization are human-readable with mixed real words and gibberish (non-word token sequences).

Although captions generated by MLLM assist reconstruction, they often fail to capture detailed textures and structures, creating a consistency gap between the reconstructed and original images. This limitation hinders fidelity. To address this, we propose a differential optimization process tailored for the diffusion model. Our goal is to find the prompt that is most suitable for the entire ResULIC with the best perceptual fidelity.

Formally, input captions are first converted into token indices using a tokenizer, referencing a predefined vocabulary $E^{|V| \times d}$ used in CLIP (Radford et al., 2021), where $|V|$ is the vocabulary size of the model and $d$ is the dimension of the embeddings. Instead of random initialization as in PEZ, here we initialize the learnable continuous embedding $P = [e_i, ..., e_M]$ by previously obtained $c_{\mathbf{res}}$, where $M$ is the number of tokens worth of vectors to optimize. The subsequent optimization process can be seen in Algorithm 1.

The optimization is driven by the loss function:

$$\mathcal{L}_{\mathrm{pfo}} = \lambda_l \mathbb{E}_{z_t, \epsilon} \left[ \|\epsilon - \epsilon_\theta(z_n, n, z_c, E_{\mathrm{clip\text{-}c}}(P'))\|_2^2 \right]$$
$$+ \lambda_c \mathcal{L}_{\mathrm{aux}}(P', x), \tag{6}$$

**Algorithm 1** Pfo: Perceptual fidelity optimization

**Input:** Diffusion model: $\theta$, CLIP model: $\Omega$, Target image: $x$, Initialed embedding: $P$, Added Timesteps: $N_r$, Denoising steps: $N_d$, Selected Timesteps: $n$; Learning rate: $\lambda$, Optimization steps: $i$

1: **for** $1, ..., i$ **do**
2:    # Forward Projection:
3:    $P' \leftarrow \text{Proj}_{\mathbf{E}}(P)$
4:    # Select time step $n$:
5:    $n \leftarrow random(\{N_d/N_r, 2N_d/N_r, \dots, N_r\})$
6:    # Calculate the gradient w.r.t. the *projected* embedding:
7:    $g \leftarrow \nabla_{\mathbf{P'}}(\mathcal{L}_{\text{pro}}(\theta, \Omega, z_n, n, P', x))$
8:    # Update the embedding:
9:    $P \leftarrow P - \lambda g$
10: **end for**
11: **return** $P \leftarrow \text{Proj}_{\mathbf{E}}[P]$

---

where the first term represents the denoising loss for predicting the noise at timestep $n$, to stabilize the optimization process, we also incorporate Equation $\mathcal{L}_{\text{aux}}(P', x) = 1 - S(E_{\text{clip-c}}(P'), E_{\text{clip-v}}(x))$ as an auxiliary loss function with a small weighting factor $\lambda_c$, $E_{\text{clip-c}}$ and $E_{\text{clip-v}}$ denote the text and visual encoders of the CLIP model, $S$ is the cosine similarity between two embedding vectors. During optimization, the embedding $P$ is projected into the discrete space using the $\text{Proj}_{\mathbf{E}}$ function, which finds the nearest embedding $P' = \text{Proj}_{\mathbf{E}}(P)$ in the CLIP embedding space. Euclidean distance is used for this projection, ensuring that the learned embeddings stay aligned with the vocabulary space of the CLIP model. The final optimized texts can be obtained from the updated $P = P - \gamma \frac{d\mathcal{L}_{\text{pro}}}{dP'}$ after several iterations with learning rate $\gamma$.

**Index Coding.** In addition, we propose an efficient text encoding method that replaces zlib-based character encoding by encoding text embedding indices, with the decoding side using CLIP's text encoder to map these indices back to embeddings, significantly reducing bitrate consumption. See Appendix A.3 for details.

# 5. Compression-aware Diffusion Model (CDM)

During the forward process for typical diffusion models, Gaussian noise is gradually added to the clean latent feature $z_0$. The intensity of the noise added at each step is controlled by the noise schedule $\beta_t$. This process can be written as:

$$z_t = \sqrt{\bar{\alpha}_t} z_0 + \sqrt{1 - \bar{\alpha}_t}\epsilon, \quad t \in \{1, 2, \dots, T\}, \quad (7)$$

where $\epsilon \sim \mathcal{N}(0, \mathbf{I})$ is a sample from a standard Gaussian distribution. Here, $\alpha_t = 1 - \beta_t$ and $\bar{\alpha}_t = \prod_{i=1}^{t} \alpha_i$. As $t$ increases, the corrupted representation $z_t$ gradually approaches a Gaussian distribution.

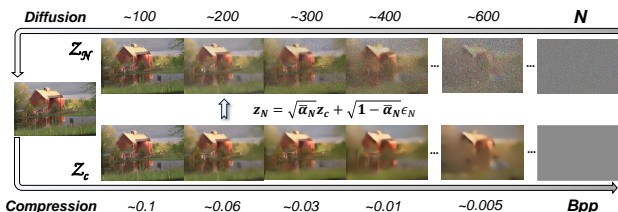

*Figure 6.* The intrinsic relationship between diffusion steps and compression ratios (bitrates). Larger diffusion steps correspond to lower bitrates.

In image compression, efficiency and reliability take precedence over the diversity of reconstructed images. Starting from standard Gaussian noise introduces unnecessary uncertainty. Therefore, we aim to avoid scenarios where the endpoint of noise addition (i.e., the starting point of denoising) is purely random noise. To address this, we define the following formulation:

$$z_N = \sqrt{\bar{\alpha}_{N_r}} z_c + \sqrt{1 - \bar{\alpha}_{N_r}}\epsilon, \quad N_r < T. \quad (8)$$

Existing works (Yang et al., 2024; Wang et al., 2024a) employ similar strategies for tasks like deblurring and super-resolution typically, assuming fixed degradation levels (e.g., upsampling ratios). In contrast, we jointly explore the correlation among **bitrate**, **distortion**, and **diffusion steps**, as shown in Figure 7. A significant challenge lies in understanding how varying levels of compression noise, quantified by the compressed latent residual $\rho_{res} = z_c - z_0$, impact the diffusion process.

## 5.1. Modeling Diffusion Steps with Compression Levels

DDPM (Ho et al., 2020) already provided a very preliminary prototype on modeling the bitrates and the reconstruction quality under the scenario of lossy image compression. In diffusion models, the information entropy $R_N$ at each timestep $N$ can be approximately quantified using the KL divergence between the true posterior $q(z_{N-1}|z_N, z_0)$ and the model posterior $p_\theta(z_{N-1}|z_N)$ reflects the information entropy of $z_N$ as:

$$R_N \sim D_{\text{KL}}(q(z_{N-1} \mid z_N, z_0) \, \| \, p_\theta(z_{N-1} \mid z_N)). \quad (9)$$

This brutal correlation mapping between Rate and Diffusion steps provides the intuition that: At larger $N$ (near the end of the diffusion process with high noise), $z_N$ carries less information about $z_0$. Thus, the KL divergence and $R_N$ are smaller, enabling higher compression rates but lower reconstruction quality. In contrast, at smaller $N$ (near the start of the diffusion process with low noise), $z_N$ retains more information about $z_0$, leading to larger KL divergence and $R_N$, resulting in lower compression rates but higher reconstruction quality.

Figure 6 visually illustrates this principle. To address these variations, we need to dynamically adjust $N$ based on the

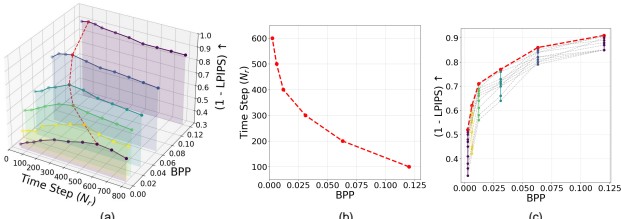

*Figure 7.* **Correlation between *Bitrates* and *Diffusion Steps*.** (a) Reconstruction quality reaches the peak at different diffusion steps for different bitrates. (b) **Peak Projection Curve** where optimal $N_r$ decreases as bpp increases. (c) The adaptive strategy in (b) performs notably better (red curve) than fixed diffusion steps.

compression ratio. In Figure 7, we present a three-factor analysis of rate (bpp), $N_r$, and quality (LPIPS). As observed, reconstruction quality varies with bitrate. Fixing the denoising steps, as shown in Figure 7(c), results in suboptimal performance. Guided by the observation in Figure 7, we propose to adapt the fixed $N$ into a compressed bitrate-adaptive $N_r$. Specifically, for different compression ratios, our method selects varying noise-adding endpoints while retaining the original noise schedule of the diffusion model. CDM overcomes the limitation of modifying noise schedules in the pre-trained models, achieving visually pleasing reconstruction without the need for retraining. Details of the diffusion process is provided in the following parts. Relevant proofs and derivations are provided in the subsequent sections and the Appendix B.

### 5.2. Noise Adding Process

Existing methods that utilize residual diffusion, such as ResShift (Yue et al., 2024) and RDDM (Liu et al., 2024b), typically employ new noise schedules and design a Markovian forward process. Here, to implement Equation (8), we adopt the noise schedule of Stable Diffusion (Rombach et al., 2022) and design a non-Markovian forward process that does not rely on $q(z_n|z_{n-1})$.

**Definition 5.1** (*Noise Addition Mechanism*).

$$q(z_n|z_0, \rho_{\text{res}}) \sim \mathcal{N}\left(\sqrt{\bar{\alpha}_n}z_0 + \sqrt{1-\bar{\alpha}_n}\gamma_n\rho_{\text{res}}, (1-\bar{\alpha}_n)\mathbf{I}\right)$$
(10)

where $z_n$ can be sampled using the following equation:

$$z_n = \sqrt{\bar{\alpha}_n}z_0 + \sqrt{1-\bar{\alpha}_n}\left(\gamma_n\rho_{\text{res}} + \epsilon_n\right), \ n \in \{1, \dots, N_r\}$$
(11)

### 5.3. Reverse Sampling Process

**Theorem 5.2** (Conditional Independence of $z_n$ and $z_{n-1}$). *Given the distributions defined in Equation (11), we have*
$$z_n \perp z_{n-1} \mid z_0, z_c,$$

*Proof.* Under the given model, $z_n$ is determined solely by $(z_0, z_c)$ and the noise term $\epsilon_n$, which is independent of

$\epsilon_{n-1}$. Consequently, conditioning on $z_{n-1}$ provides no additional information about $z_n$, implying $q(z_n|z_{n-1}, z_0, z_c) = q(z_n|z_0, z_c)$. Hence, $z_n \perp z_{n-1} \mid z_0, z_c$. □

Based on Theorem 5.2, the conditional probability simplifies to $q(z_{n-1}|z_n, z_0, z_c) = q(z_{n-1}|z_0, z_c) \sim \mathcal{N}\left(\sqrt{\bar{\alpha}_{n-1}}z_0 + \sqrt{1-\bar{\alpha}_{n-1}}\gamma_{n-1}\rho_{\text{res}}, (1-\bar{\alpha}_{n-1})\mathbf{I}\right)$.

**Assumption 5.3.** *Assume that the conditional probability can also be expressed as:*

$$q(z_{n-1} \mid z_n, z_0, z_c) \sim \mathcal{N}\left(z_{n-1}; \iota_n z_n + \zeta_n z_0, \sigma_n^2\mathbf{I}\right).$$
(12)

Substituting Equation (11) into Equation (12), we obtain:

$$\iota_n = \frac{\sqrt{1-\bar{\alpha}_{n-1}}}{\sqrt{1-\bar{\alpha}_n}},$$
(13)

$$\zeta_n = \sqrt{\bar{\alpha}_{n-1}} - \sqrt{\bar{\alpha}_n} \cdot \iota_n,$$
(14)

$$\gamma_n = \gamma_{n-1} = \frac{\sqrt{\bar{\alpha}_{N_r}}}{\sqrt{1-\bar{\alpha}_{N_r}}}.$$
(15)

Here, we set $\sigma_n$ to the same as in DDIM (Song et al., 2021), $\sigma_n = \eta\sqrt{\frac{1-\bar{\alpha}_{n-1}}{1-\bar{\alpha}_n}}\sqrt{1 - \frac{\bar{\alpha}_n}{\bar{\alpha}_{n-1}}}$ where $\eta$ is a hyperparameter. By setting $\eta = 0$ or $\eta = 1$, we correspond to different sampling strategies, i.e., deterministic sampling or stochastic sampling. In the following experiments, $\eta$ is set to 0 unless stated otherwise. Comprehensive derivations and experiments are available in Appendix B.5 and B.6. During directional sampling, samples are drawn from the model-predicted distribution $p_\theta(z_{n-1}|z_n, \tilde{z}_0, z_c)$, i.e., $z_{n-1} = \iota_n z_n + \zeta_n \tilde{z}_0 + \sigma_n\epsilon$, as illustrated in Algorithm 2 in the Appendix B.6.

### 5.4. Training Objective

While keeping the diffusion model fixed, we need to train the encoder and decoder in the latent space as well as the visual adapter. We derive the following simplified loss function for training, detailed proofs are provided in Appendix B.7:

$$\mathcal{L}_{\text{Vis}} = \omega_n^2 \cdot \mathbb{E}_{z_0, c, z_c, n, \epsilon} \left\| \epsilon - \epsilon_\theta(z_n, c, z_c, t) \right\|^2$$
(16)

where $\omega_n^2 = \left(\frac{\zeta_n\sqrt{1-\bar{\alpha}_n}}{\sqrt{\bar{\alpha}_n} - \sqrt{1-\bar{\alpha}_n}\cdot\gamma_n}\right)^2$ and $c$ is the condition from texts. During training, we omitting the $\omega_n^2$ parameter to stabilize training. Furthermore, we followed Equation (36) in the Appendix to estimate the original data $\hat{z}_0$, decode it into the pixel domain $\hat{x}_0$, and performed further perceptual optimization using a weighted combination of LPIPS and MSE. Then we can rewrite Equation (16) as:

$$\mathcal{L}_{\text{Vis}} = \mathbb{E}_{z_0, c, z_c, n, \epsilon} \left\| \epsilon - \epsilon_\theta(z_n, c, z_c, t) \right\|^2 + \lambda_d \text{MSE}(x_0, \hat{x}_0) + \lambda_p \text{LPIPS}(x_0, \hat{x}_0)$$
(17)

Combining the $\mathcal{L}_{\text{Vis}}$ with feature compressor, we define the total loss function as:

$$\mathcal{L} = \mathcal{L}_{\text{Vis}} + \mathcal{L}_D + \lambda_R\mathcal{L}_R$$
(18)

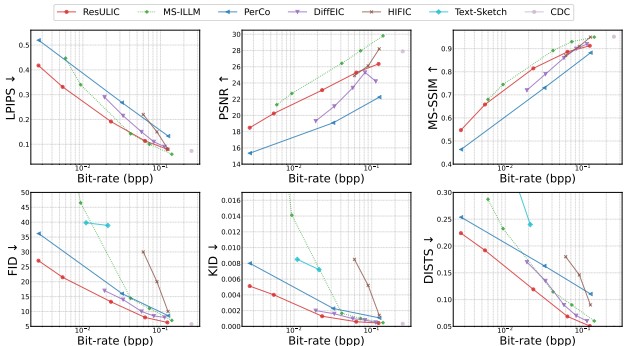

*Figure 8.* Quantitative comparision with state-of-the-art methods on CLIC-2020 dataset.

$\mathcal{L}_D = \lambda D(z_0, z_c)$ here is the distortion between $z_0$ and $z_c$. $\mathcal{L}_R = R(z_c)$ indicates the rate of the compressed latent. We adjust $\lambda_R$ to control the trade-off between rate, distortion, and the diffusion prediction loss.

## 6. Experiments

### 6.1. Experimental Setup

Stable diffusion v2.1 (Rombach et al., 2022) is used as the backbone diffusion model. And the MLLM GPT4o (OpenAI, 2024) is applied to capture the image caption and their corresponding residual information. See Appendix C.4 for training details.

For evaluation, we tested several widely used datasets including CLIC-2020 (Toderici et al., 2020) in the main paper, and Kodak (Kodak, 1993), DIV2K (Agustsson & Timofte, 2017), Tecnick (Asuni & Giachetti, 2014) and MS-COCO (Caesar et al., 2018) in Appendix A. Multiple metrics are evaluated including PSNR, MS-SSIM (Wang et al., 2003), LPIPS (Zhang et al., 2018), DISTS (Ding et al., 2020) FID (Heusel et al., 2017) and KID (Bińkowski et al., 2018). Among them, FID and KID are used to evaluate perceptual realism by matching the feature distributions between the original and reconstructed image sets. In contrast, LPIPS and DISTS balance realism and fidelity, with relatively greater emphasis on the latter.

We compare our method to state-of-the-art codecs, including the traditional codec VTM (Bross et al., 2021), GAN-based neural compressors HiFiC (Mentzer et al., 2020), MS-ILLM (Muckley et al., 2023), as well as diffusion-based compressors Text-sketch (Lei et al., 2023a), DiffEIC (Li et al., 2024a), CDC (Yang & Mandt, 2022) and PerCo (Careil et al., 2024). PerCo evaluated here is a reproduced version[1], as the official implementation is not publicly available. The diffusion model in this version has been fully refined for the compression task. Additionally, we provide comparisons with the original paper's reported results in

[1]https://github.com/Nikolai10/PerCo

Appendix D.

### 6.2. Main Results

Figure 8 presents a quantitative comparison across multiple metrics on the CLIC-2020 dataset. The results demonstrate that our method, ResULIC, consistently outperforms state-of-the-art approaches on perceptual realism metrics such as FID and KID. Moreover, for perceptual fidelity metrics like DISTS, ResULIC achieves superior performance, while for LPIPS, it excels at relatively low bitrates, highlighting its effectiveness in improving fidelity. Additional comparisons on datasets including Kodak, DIV2K, Tecnick, and MS-COCO are provided in Appendix A.1.

In addition, we performed a BD-Rate (Bjontegaard, 2001) comparison with existing methods using MS-ILLM as the anchor, as shown in Table 1. Our method achieves the best performance in terms of perceptual realism and perceptual fidelity, while outperforming existing diffusion-based ultra-low bitrate compression methods on distortion metrics.

*Table 1.* BD-Rate (%) ↓ comparison with state-of-the-art methods on CLIC-2020 datasets. **ResULIC w/o Pfo** is a faster variant with the Pfo module disabled, while still maintaining good performance.

| Type | Methods | Perceptual Realism | | Perceptual Fidelity | | Distortion | |
|---|---|---|---|---|---|---|---|
| | | FID | KID | DISTS | LPIPS | PSNR | MS-SSIM |
| **GAN based** | MS-ILLM | 0 | 0 | 0 | 0 | 0 | 0 |
| | HiFiC | 65.0 | 154.7 | 39.0 | 75.7 | 122.3 | 46.2 |
| **Diffusion based** | DiffEIC | -28.4 | -21.9 | 1.7 | 9.5 | 392.3 | 133.8 |
| | PerCo | -17.9 | -25.7 | -0.77 | 113.4 | 1092.2 | 244.1 |
| | ResULIC w/o Pfo | -54.9 | -64.7 | -45.7 | -18.8 | 120.4 | 41.8 |
| | ResULIC | **-62.9** | **-68.5** | **-51.6** | **-26.9** | 120.3 | 37.2 |

**Complexity.** The average encoding time for a 768×512 Kodak image is 0.10s for the latent compressor, 5.77s for semantic residual coding using Srr (+180s if Pfo is enabled). Decoding averages 0.60s/0.43s for 4/3 denoising steps. More detailed complexity analysis is in Appendix C.3.

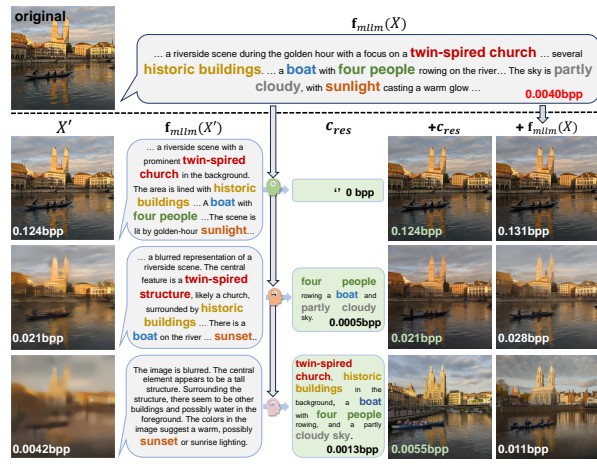

*Figure 9.* Demonstration of the **Srr** at different bitrates. As the bitrates get smaller, more residual information would be retrieved. Keywords are highlighted with colors.

*Table 2.* Ablation experiments comparing each module on the CLIC-2020 dataset using PerCo as the anchor.

| Method | w/ CDM | w/ Index Coding | w/ Srr | w/ Pfo | BD-Rate (%) ↓ | | | | | |
|---|---|---|---|---|---|---|---|---|---|---|
| | | | | | LPIPS | DISTS | FID | KID | PSNR | MS-SSIM |
| | – | × | – | × | 0 | 0 | 0 | 0 | 0 | 0 |
| PerCo | – | ✓ | – | × | -6.7 | -6.6 | -6.8 | -7.7 | -6.1 | -6.6 |
| | – | ✓ | – | ✓ | -15.9 | -14.6 | -15.7 | -14.1 | -5.9 | -7.0 |
| ResULIC-VQ | – | × | – | × | 2.8 | -2.1 | -10.5 | -9.6 | -36.0 | -18.8 |
| | × | × | × | × | -15.6 | -12.3 | -15.1 | -13.8 | -74.4 | -22.7 |
| | ✓ | × | × | × | -53.1 | -46.9 | -36.1 | -38.4 | -76.2 | -48.8 |
| ResULIC | ✓ | ✓ | × | × | -60.7 | -58.1 | -46.7 | -48.4 | -87.0 | -59.0 |
| | ✓ | ✓ | ✓ | × | -71.6 | -68.2 | -57.9 | -59.1 | **-96.8** | -69.1 |
| | ✓ | ✓ | ✓ | ✓ | **-80.7** | **-77.3** | **-66.3** | **-66.8** | -96.2 | **-70.2** |

*Table 3.* Performance comparison of MLLM models on Kodak.

| Method | MLLM | BD-Rate (%) ↓ | | | |
|---|---|---|---|---|---|
| | | LPIPS | DISTS | PSNR | MS-SSIM |
| **ResULIC w/o Pfo** | GPT-4o | 0 | 0 | 0 | 0 |
| | Llama-3.2-11B | 1.5 | 0.1 | 1.5 | -0.2 |
| | Qwen-VL-MAX | **-1.6** | 3.2 | 0.5 | 0.52 |
| | SenseChat-Vision | 0.42 | -1.1 | 0.3 | 0.8 |
| **ResULIC** | GPT-4o | -6.1 | -3.8 | 0.8 | 3.1 |
| | Llama-3.2-11B | −5.8 | -4.2 | 1.2 | 3.5 |
| | Qwen-VL-MAX | -6.0 | -3.9 | 0.9 | 3.1 |
| | SenseChat-Vision | -5.7 | -4.0 | 1.1 | 3.4 |

### 6.3. Evaluation

Table 2 presents a comprehensive ablation study on the performance impact of each proposed module.

**Case 1: Detailed comparison with PerCo.** To more clearly demonstrate the advantages of our method over PerCo, we used PerCo as an anchor for comparison. (1) The results in Table 2 demonstrate that the Pfo module and Index Coning integrate seamlessly into the PerCo framework, consistently improving reconstruction quality. (2) To solely evaluate the impact of the latent compressor, we provide a special version, ResULIC-VQ, which replaces our latent compressor with the same VQ compressor used in PerCo and disables all newly introduced modules. These comparisons highlight that the stable gains originate from our proposed modules.

**Case 2: Effectiveness of Semantic Residual Coding.**

To quantify the gains of **SRC**, we conducted comprehensive experiments on the CLIC-2020 dataset. The results in Table 2 demonstrate that the Srr and Pfo module brings consistent improvements in realism and fidelity to ResULIC.

**Adaptiveness of Srr to different bitrates.** In Figure 9, as the bitrate decreases, semantic information in $x'$ diminishes, leading to vague or incorrect descriptions, yet reconstruction quality remains stable. Figure 10 further shows that as total bpp decreases, the text bitrate proportion increases significantly, highlighting Srr's adaptivity at varying bitrates. Additional visual examples are provided in the Appendix D.

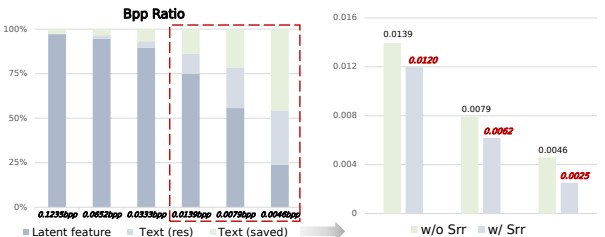

*Figure 10.* The relative text/latent bitrate ratio under different total bitrates. The **Srr** effectively removes more redundancy at lower bitrates.

**The impact of different MLLMs.** MLLM models have been evolving rapidly. Existing methods may use different MLLMs for text retrieval. Here, we evaluate several models, including one offline model Llama-3.2-11B-Vision-

Instruct (Dubey et al., 2024), and three online models gpt-4o (OpenAI, 2024), SenseChat-Vision, and Qwen-VL-MAX (Wang et al., 2024b). Table 3 demonstrates the flexibility of our method across different MLLMs. The tested MLLMs achieve stable and comparable performance, especially after applying the Pfo module to optimize semantic retrieval.

**Case 3: Effectiveness of Compression-aware Diffusion.**

By integrating *Definition* 5.1, ResULIC leverages the existing noise schedule of Stable Diffusion. This approach eliminates the computational overhead associated with retraining a new latent diffusion model, as required by methods like ResShift and RDDM when modifying the noise schedule. It also significantly reduces denoising time redundancy while achieving substantial improvements across all metrics.

Table 4 uses the Adjustable Noise Schedule (ANS) proposed in PASD (Yang et al., 2024) as the baseline for comparison. ANS affects only the inference stage by introducing signal information from the low-quality input image. Compared to w/o CDM (i.e., without using our *Definition* 5.1 for training, and initializing from random noise during testing), ANS shows some improvements, but it remains inadequate. In contrast, our proposed CDM bridges training and inference, effectively reducing the train-test discrepancy and achieving superior performance.

*Table 4.* Bitrate-aware Diffusion enables both accelerated inference speed (with fewer steps) and improved BD-rate performance.

| Method | w/ CDM (Denoising Steps) | BD-Rate(%) ↓ | | | |
|---|---|---|---|---|---|
| | | LPIPS | DISTS | PSNR | MS-SSIM |
| ANS | ×(50/20) | 0 | 0 | 0 | 0 |
| ResULIC | ×(50/20) | 4.6 | 5.3 | 6.8 | 47.1 |
| | ✓ (4/3) | -54.4 | -65.8 | -49.7 | -23.9 |

## 7. Conclusion

In this paper, we proposed ResULIC, a residual-guided ultra lowrate image compression with Semantic Residual Coding and Biterate-aware Diffusion. Extensive experiments show both promising perceptual quality and reliability under ultra low bitrates. The proposed optimization strategy also demonstrates flexibility in being quickly applied to existing frameworks.

## Acknowledgement

This work was supported in part by the Key Project of Jiangsu Science and Technology Department under Grant BK20243038, and in part by the Key Project of the National Natural Science Foundation of China under Grant 62431011. The authors would like to express their sincere gratitude to the Interdisciplinary Research Center for Future Intelligent Chips (Chip-X) and Yachen Foundation for their invaluable support. This project was also funded by Kuaishou Technology.

## Impact Statement

This work aims to advance the field of machine learning with applications in image compression. It holds potential value for research on image compression tasks under extreme bandwidth constraints and, to the best of our knowledge, raises no ethical concerns.

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

# A. Additional Evaluation

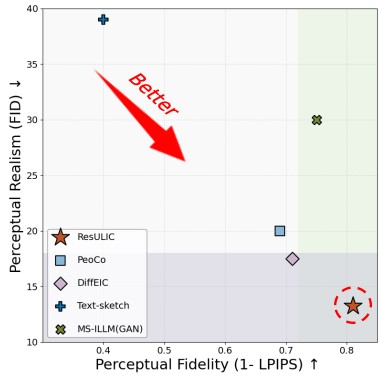

*Figure 11.* **Perceptual realism *vs.* perceptual fidelity** at 0.02 bpp on the CLIC2020 dataset.

## A.1. Comparison on other dataset

In addition to the experiments on the CLIC2020 (Toderici et al., 2020) dataset, we conducted more detailed comparisons on MS-COCO (Caesar et al., 2018), DIV2K (Agustsson & Timofte, 2017), Tecnick (Asuni & Giachetti, 2014), and Kodak (Kodak, 1993). As shown in Figure 12, 13, 14, our method consistently outperforms state-of-the-art approaches on perceptual realism metrics such as FID and KID, and achieves superior performance in perceptual fidelity metrics like DISTS, while excelling in LPIPS at relatively low bitrates, demonstrating its effectiveness in improving fidelity. Since CorrDiff (Ma et al., 2024) and GLC (Jia et al., 2024) are not open-source and tested differently on other datasets, we selected two bitrate points from their Kodak experiments for comparison, as illustrated in Figure 15.

**Evaluation details**: For evaluation on the CLIC2020, DIV2K, and Tecnick datasets, we followed the approach of CDC (Yang & Mandt, 2022) by resizing images to a short side of 768 and then center-cropping them to 768×768. For MSCOCO-3K, we randomly selected 3,000 images from the MSCOCO dataset and, following PerCo, resized them to 512×512 for testing. The FID and KID metrics for all datasets were calculated on 256×256 patches, as described in HiFiC (Mentzer et al., 2020).

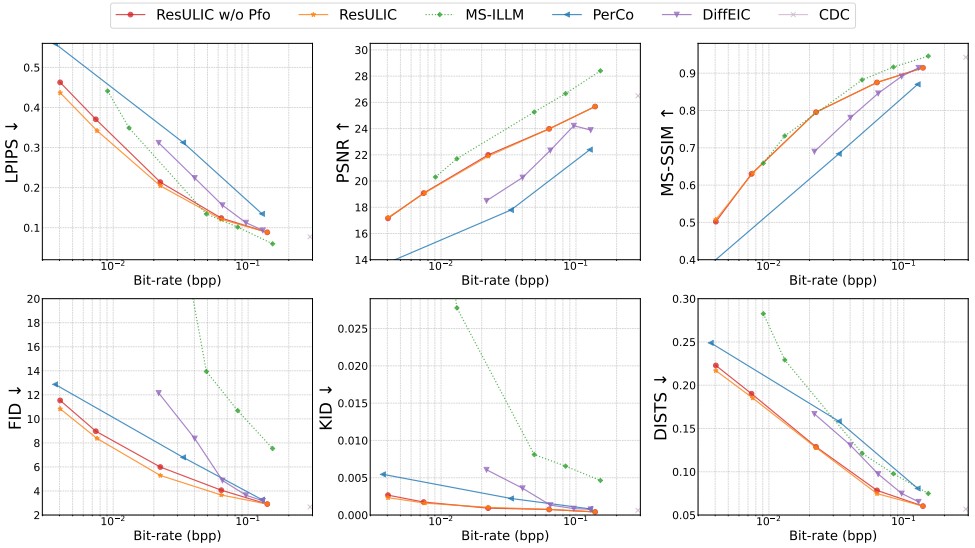

*Figure 12.* Quantitative comparision with state-of-the-art methods on MSCOCO-3K datasets.

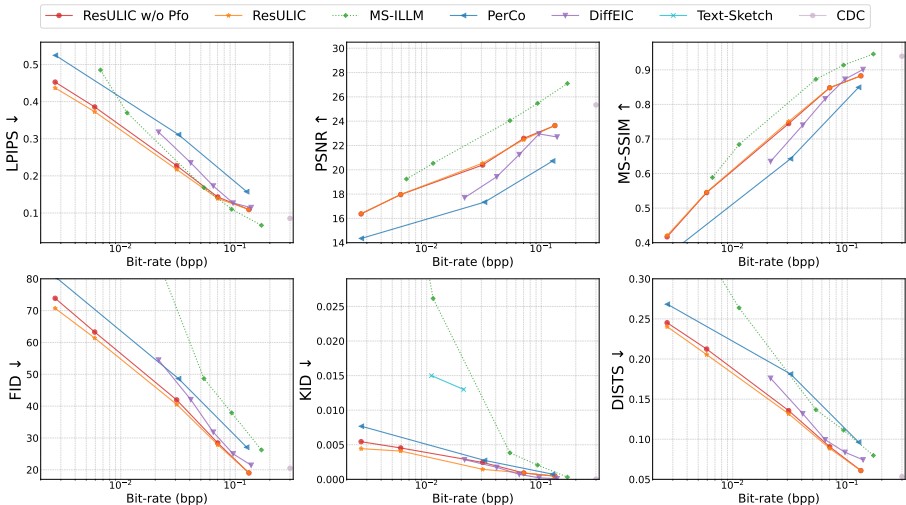

*Figure 13.* Quantitative comparision with state-of-the-art methods on DIV2K datasets.

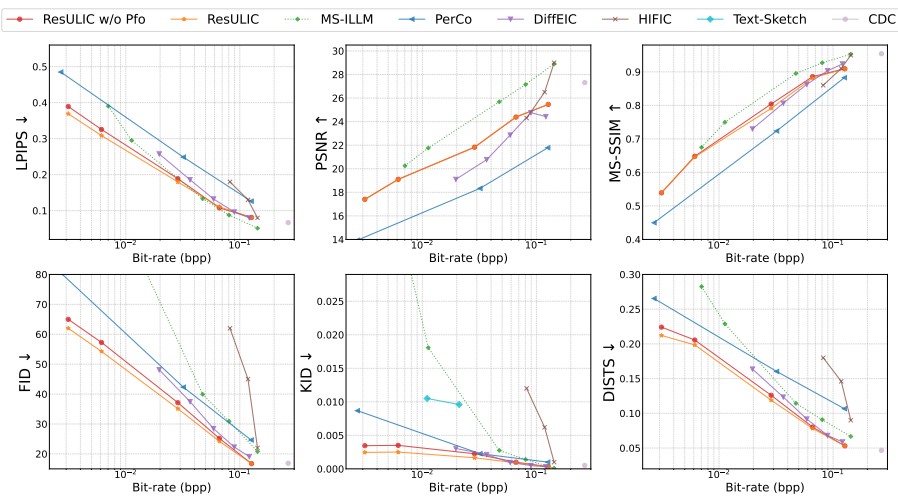

*Figure 14.* Quantitative comparison with state-of-the-art methods on Tecnick datasets.

## A.2. Additional Experiments of Pfo

We also compared our method with existing approaches, such as the hard prompt method PEZ. Both visual and quantitative results confirm that Pfo significantly enhances perceptual quality and objective metric performance, particularly under low-bitrate conditions. For example, as shown in Table 5, the optimized textual information produced by Pfo improves the fidelity of frameworks like PerCo and Text+Sketch, which employ entirely different architectures for compressing low-level content. Notably, this enhancement is achieved without requiring additional retraining, as the Pfo module leverages pretrained MLLM models, making it a flexible and efficient plug-in tool.

## A.3. The Impact of Index Coding

Our proposed index coding is a simple yet effective method for text encoding. As illustrated in Figure 16, the text encoding logic of CLIP first tokenizes the prompt into several words, matches these words to their corresponding indices in a predefined vocabulary, and then maps these indices to their respective text embedding vectors. In contrast to directly encoding text strings, our method encodes the indices corresponding to CLIP's text embedding vectors. From Figure 16, we can see that our method can save a lot of text bit rate consumption compared to zlib.

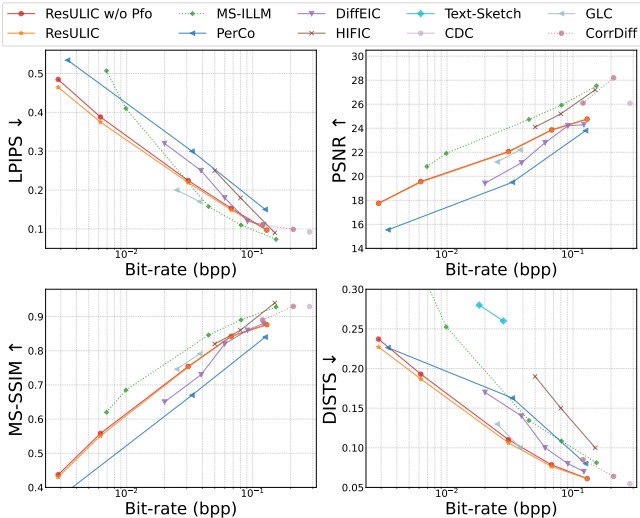

*Figure 15.* Quantitative comparison with state-of-the-art methods on Kodak datasets.

*Table 5.* Evaluation of Pfo with different low-level content compressors. PerCO uses a "codebook+hyperprior" model for compression. Text+Sketch compresses the binary contour maps and has already applied PEZ in its framework.

| Compressor | bpp/LPIPS ↓ | | |
|---|---|---|---|
| (content) | *No optimizer* | w/ PEZ | **w/ Pfo** |
| PerCo | 0.0033/0.546 | 0.0035/0.532 | 0.0033/**0.518** |
| Text+Sketch | - | 0.0280/0.586 | 0.0280/**0.565** |
| ResULIC | 0.0028/0.49 | 0.0028/0.49 | 0.0028/**0.46** |

## A.4. The impact of sampling method

As shown in Appendix B.5, when $\eta = 0$ and $\eta = 1$, the sampling process corresponds to deterministic sampling and stochastic sampling, respectively. We conducted experiments on the CLIC2020 dataset, with the results presented in the figure below. These results reveal that adding noise during sampling compromises the consistency of reconstruction. Therefore, deterministic sampling ($\eta = 0$) is the better choice.

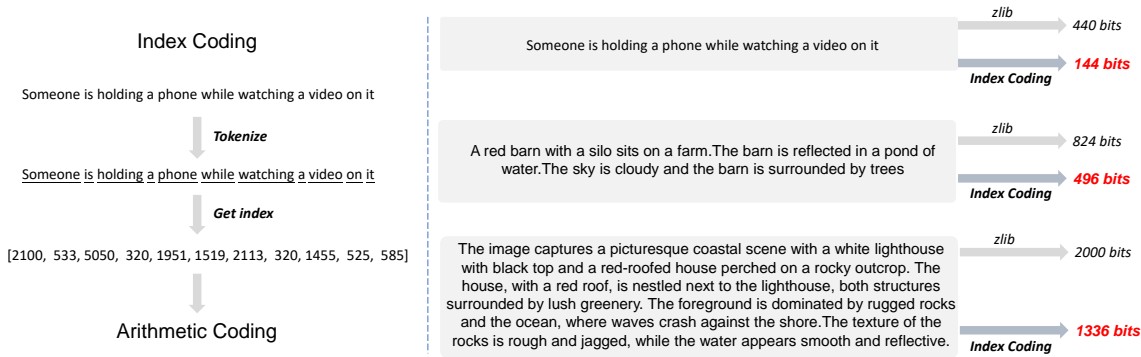

*Figure 16.* Visualization of Index Coding.

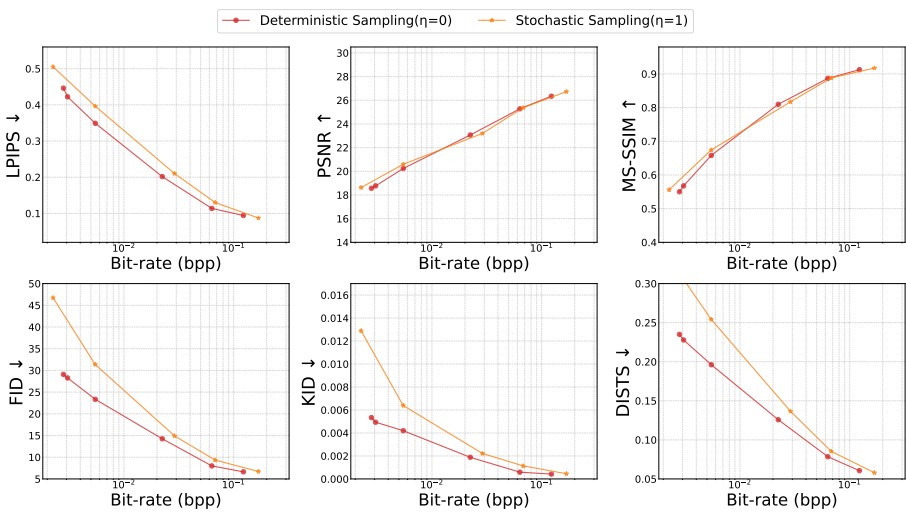

*Figure 17.* Quantitative comparision of deterministic sampling($\eta = 0$) and stochastic sampling($\eta = 1$) on CLIC2020 datasets.

## B. Mathematical Framework

In this section, we formalize the noise addition mechanism integral to our model and derive the essential parameter relationships. We begin by defining the distributions of the latent variables $z_n$ and $z_{n-1}$, establish their independence, and subsequently derive the conditional probabilities. Finally, we present the parameter settings for different sampling methods, specifically DDIM and DDPM, and discuss the formulation of the training loss.

### B.1. Noise Addition Mechanism

**Definition B.1** (Noise Addition Mechanism). For each timestep $n$, the latent variables $z_n$ and $z_{n-1}$ are defined by the following Gaussian distributions:

$$z_n \sim \mathcal{N}\left(\sqrt{\bar{\alpha}_n}z_0 + \sqrt{1 - \bar{\alpha}_n}\gamma_n\rho_{\text{res}},\ (1 - \bar{\alpha}_n)\mathbf{I}\right), \tag{19}$$

$$z_{n-1} \sim \mathcal{N}\left(\sqrt{\bar{\alpha}_{n-1}}z_0 + \sqrt{1 - \bar{\alpha}_{n-1}}\gamma_{n-1}\rho_{\text{res}},\ (1 - \bar{\alpha}_{n-1})\mathbf{I}\right). \tag{20}$$

Here, $\bar{\alpha}_n$ denotes the cumulative product of $\alpha_n$ up to timestep $n$, $\gamma_n$ is a scaling factor, $\rho_{\text{res}}$ represents a residual term, and $\mathbf{I}$ is the identity matrix.

## B.2. Independence of Consecutive Latent Variables

**Theorem B.2** (Conditional Independence of $z_n$ and $z_{n-1}$)**.** *Given the distributions defined in Equations* (11)*, we have*
$$z_n \perp z_{n-1} \mid z_0, z_c,$$

*Proof.* Under the given model, $z_n$ is determined solely by $(z_0, z_c)$ and the noise term $\epsilon_n$, which is independent of $\epsilon_{n-1}$. Consequently, conditioning on $z_{n-1}$ provides no additional information about $z_n$, implying $q(z_n|z_{n-1}, z_0, z_c) = q(z_n|z_0, z_c)$. Hence, $z_n \perp z_{n-1} \mid z_0, z_c$. $\qquad\square$

## B.3. Conditional Probability Reduction

Based on Theorem B.2, the conditional probability simplifies to:
$$q(z_{n-1} \mid z_n, z_0, z_c) = q(z_{n-1} \mid z_0, z_c) \sim \mathcal{N}\left(\sqrt{\bar{\alpha}_{n-1}}z_0 + \sqrt{1 - \bar{\alpha}_{n-1}}\gamma_{n-1}\rho_{\text{res}}, \ (1 - \bar{\alpha}_{n-1})\mathbf{I}\right). \tag{21}$$

## B.4. Derivation of Parameter Relationships

Assume that the conditional probability can also be expressed as:
$$q(z_{n-1} \mid z_n, z_0, z_c) \sim \mathcal{N}\left(z_{n-1}; \ \iota_n z_n + \zeta_n z_0, \ \sigma_n^2 \mathbf{I}\right). \tag{22}$$

**Theorem B.3** (Parameter Relationships)**.** *For Equations* (21) *and* (22) *to be equivalent, the following system of equations must be satisfied:*
$$\begin{cases} \iota_n\sqrt{\bar{\alpha}_n} + \zeta_n = \sqrt{\bar{\alpha}_{n-1}}, \\ \iota_n^2(1 - \bar{\alpha}_n) + \sigma_n^2 = 1 - \bar{\alpha}_{n-1}, \\ \sqrt{1 - \bar{\alpha}_{n-1}}\gamma_{n-1} = \iota_n\gamma_n\sqrt{1 - \bar{\alpha}_n}. \end{cases} \tag{23}$$

*Proof.* Equate the expressions for $z_{n-1}$ from Equations (21) and (22):
$$z_{n-1} = \iota_n z_n + \zeta_n z_0 + \sigma_n \epsilon, \tag{24}$$
$$z_{n-1} = \sqrt{\bar{\alpha}_{n-1}}z_0 + \sqrt{1 - \bar{\alpha}_{n-1}}\left(\gamma_{n-1}\rho_{\text{res}} + \epsilon_{n-1}\right). \tag{25}$$

Substitute the expression for $z_n$ from Equation (19) into the first equation:
$$\begin{aligned} z_{n-1} &= \iota_n\left(\sqrt{\bar{\alpha}_n}z_0 + \sqrt{1 - \bar{\alpha}_n}\gamma_n\rho_{\text{res}} + \sqrt{1 - \bar{\alpha}_n}\epsilon_n\right) + \zeta_n z_0 + \sigma_n\epsilon \\ &= \left(\iota_n\sqrt{\bar{\alpha}_n} + \zeta_n\right)z_0 + \iota_n\sqrt{1 - \bar{\alpha}_n}\gamma_n\rho_{\text{res}} + \left(\sigma_n\epsilon + \iota_n\sqrt{1 - \bar{\alpha}_n}\epsilon_n\right). \end{aligned} \tag{26}$$

Comparing coefficients with the second expression for $z_{n-1}$, we obtain the system of equations in (23). The noise terms $\epsilon$ and $\epsilon_n$ are mutually independent and follow standard normal distributions, leading to the condition on the variances. $\qquad\square$

## B.5. Sampling Method Parameter Settings

The parameter $\sigma_n$ varies depending on the chosen sample method. We present the parameter settings for two prominent sampling methods: **Deterministic Sampling** and **Stochastic Sampling**.

### B.5.1. DETERMINISTIC SAMPLING

When employing the **Deterministic Sampling**, the noise parameter is set to zero:
$$\sigma_n = 0. \tag{27}$$

Substituting $\sigma_n = 0$ into the system of equations (23), we derive the specific parameter configurations for the sampler:
$$\iota_n = \frac{\sqrt{1 - \bar{\alpha}_{n-1}}}{\sqrt{1 - \bar{\alpha}_n}}, \tag{28}$$
$$\zeta_n = \sqrt{\bar{\alpha}_{n-1}} - \sqrt{\bar{\alpha}_n} \cdot \frac{\sqrt{1 - \bar{\alpha}_{n-1}}}{\sqrt{1 - \bar{\alpha}_n}}, \tag{29}$$
$$\gamma_n = \gamma_{n-1}. \tag{30}$$

To ensure consistency with Equation (19), we set:

$$\gamma_n = \frac{\sqrt{\bar{\alpha}_{N_r}}}{\sqrt{1 - \bar{\alpha}_{N_r}}}.$$

### B.5.2. STOCHASTIC SAMPLING

For the Stochastic Sampling, the noise parameter is defined as:

$$\sigma_n = \sqrt{\frac{1 - \bar{\alpha}_{n-1}}{1 - \bar{\alpha}_n}} \cdot \sqrt{1 - \frac{\bar{\alpha}_n}{\bar{\alpha}_{n-1}}}. \tag{31}$$

Substituting this $\sigma_n$ into the system of equations (23), we obtain the parameter settings for the sampler:

$$\iota_n = \frac{1 - \bar{\alpha}_{n-1}}{1 - \bar{\alpha}_n} \cdot \frac{\sqrt{\bar{\alpha}_n}}{\sqrt{\bar{\alpha}_{n-1}}}, \tag{32}$$

$$\zeta_n = \sqrt{\bar{\alpha}_{n-1}} - \sqrt{\bar{\alpha}_n} \cdot \left( \frac{1 - \bar{\alpha}_{n-1}}{1 - \bar{\alpha}_n} \cdot \frac{\sqrt{\bar{\alpha}_n}}{\sqrt{\bar{\alpha}_{n-1}}} \right), \tag{33}$$

$$\gamma_n = \left( \frac{\sqrt{1 - \bar{\alpha}_n}}{\sqrt{\bar{\alpha}_n}} \Big/ \frac{\sqrt{1 - \bar{\alpha}_n - 1}}{\sqrt{\bar{\alpha}_n - 1}} \right) \cdot \gamma_{n-1}. \tag{34}$$

To maintain consistency with Equation (19), we set:

$$\gamma_n = \frac{\sqrt{1 - \bar{\alpha}_n}}{\sqrt{\bar{\alpha}_n}} \cdot \left( \frac{\bar{\alpha}_{N_r}}{1 - \bar{\alpha}_{N_r}} \right).$$

## B.6. Sampling

---
**Algorithm 2** Compression-aware Diffusion (Sampling)

---
**Input:** Diffusion model $\theta$, compressed feature $z_c$
1: Compute $z_{N_r} = \sqrt{\bar{\alpha}_{N_r}} z_c + \sqrt{1 - \bar{\alpha}_{N_r}} \epsilon_{N_r}$
2: **for** $n = N_r, \cdots, 1$ **do**
3:     Compute $\tilde{z}_0$ based on $n$:
4:     **if** $n = N_r$ **then**
5:         $\tilde{z}_0 \leftarrow z_c$
6:     **else**
7:         $\tilde{z}_0 \leftarrow \frac{\sqrt{\bar{\alpha}_n} z_n - \sqrt{1 - \bar{\alpha}_n}(\gamma_n z_c + \tilde{\epsilon}_n)}{\sqrt{\bar{\alpha}_n} - \sqrt{1 - \bar{\alpha}_n} \gamma_n}$
8:     **end if**
9:     Compute $\iota_n, \zeta_n, \sigma_n$ by $\iota_n = \frac{\sqrt{1 - \alpha_n}}{\sqrt{\alpha_n}}, \zeta_n = \sqrt{\alpha_{n-1}} - \frac{\alpha_n}{\sqrt{\alpha_{n-1}}}, \sigma_n = \eta \sqrt{\frac{1 - \bar{\alpha}_{n-1}}{1 - \bar{\alpha}_n}} \sqrt{1 - \frac{\bar{\alpha}_n}{\bar{\alpha}_{n-1}}}$,
10:     Compute $z_{n-1} = \iota_n z_n + \zeta_n \tilde{z}_0 + \sigma_n \epsilon$
11: **end for**
12: **return** $z_0$

---

## B.7. Training Objective

For training, based on Equation (19), we can express $z_n$ as:

$$
\begin{aligned}
z_n &= \sqrt{\bar{\alpha}_n} z_0 + \sqrt{1 - \bar{\alpha}_n} \left( \gamma_n \rho_{\text{res}} + \epsilon_n \right) \\
&= \sqrt{\bar{\alpha}_n} z_0 + \sqrt{1 - \bar{\alpha}_n} \gamma_n \rho_{\text{res}} + \sqrt{1 - \bar{\alpha}_n} \epsilon_n \\
&= \sqrt{\bar{\alpha}_n} z_0 + \sqrt{1 - \bar{\alpha}_n} \gamma_n (z_c - z_0) + \sqrt{1 - \bar{\alpha}_n} \epsilon_n \\
&= \left( \sqrt{\bar{\alpha}_n} - \sqrt{1 - \bar{\alpha}_n} \gamma_n \right) z_0 + \sqrt{1 - \bar{\alpha}_n} \gamma_n z_c + \sqrt{1 - \bar{\alpha}_n} \epsilon_n.
\end{aligned} \tag{35}
$$

From Equation (35), we derive the expression for $z_0$ as follows:

$$z_0 = \begin{cases} \dfrac{1}{\sqrt{\bar{\alpha}_n} - \sqrt{1-\bar{\alpha}_n} \cdot \gamma_n} \left(z_n - \sqrt{1-\bar{\alpha}_n} \cdot (\gamma_n z_c + \epsilon_n)\right), & \text{if } n \neq N, \\ z_c, & \text{if } n = N. \end{cases} \tag{36}$$

The optimization for visual adapter $\theta$ is achieved by minimizing the following negative ELBO, i.e.,

$$\sum_n D_{\text{KL}} \left[q(z_{n-1} \mid z_n, z_0, z_c) \parallel p_\theta(z_{n-1} \mid z_n, z_c)\right], \tag{37}$$

By combining this with Equation (22), we derive the visual loss function as follows:

$$\mathcal{L}_{\text{Vis}} = \mathbb{E}_{z_0,c,z_c,t,\epsilon} \left\| \iota_n z_n + \zeta_n z_0 - (\iota_n z_n + \zeta_n \hat{z}_0)\right\|^2,$$

Thus, for $n \neq N$, the training loss $\mathcal{L}_{\text{Vis}}$ can be reformulated as:

$$\begin{aligned} \mathcal{L}_{\text{Vis}} &= \zeta_n^2 * \mathbb{E}_{z_0,c,z_c,t,\epsilon} \|z_0 - \hat{z}_0\|^2 \\ &= \zeta_n^2 * \mathbb{E}_{z_0,c,z_c,t,\epsilon} \left\| \frac{1}{\sqrt{\bar{\alpha}_n} - \sqrt{1-\bar{\alpha}_n} \cdot \gamma_n} \left(z_n - \sqrt{1-\bar{\alpha}_n} \cdot (\gamma_n z_c + \epsilon_n)\right) \right. \\ &\quad \left. - \frac{1}{\sqrt{\bar{\alpha}_n} - \sqrt{1-\bar{\alpha}_n} \cdot \gamma_n} \left(z_n - \sqrt{1-\bar{\alpha}_n} \cdot (\gamma_n z_c + \epsilon_\theta(z_n, c, z_c, t))\right) \right\|^2 \\ &= \left( \frac{\zeta_n \sqrt{1-\bar{\alpha}_n}}{\sqrt{\bar{\alpha}_n} - \sqrt{1-\bar{\alpha}_n} \cdot \gamma_n} \right)^2 \mathbb{E}_{z_0,c,\hat{z}_c,t,\epsilon} \|\epsilon - \epsilon_\theta(z_n, c, z_c, t)\|^2. \end{aligned} \tag{38}$$

When $n = N$, the training loss $\mathcal{L}_{\text{Vis}}$ is set to zero. We define the coefficient $\omega_n^2 = \left( \frac{\zeta_n \sqrt{1-\bar{\alpha}_n}}{\sqrt{\bar{\alpha}_n} - \sqrt{1-\bar{\alpha}_n} \cdot \gamma_n} \right)^2$. To enhance training stability, this coefficient $\omega_n$ is omitted during training.

### B.8. Conclusion

By formalizing the noise addition mechanism and deriving the requisite parameter relationships, we establish a robust mathematical foundation for our model. The distinct parameter settings for deterministic and stochastic samplers facilitate flexibility in sampling strategies while maintaining consistency with the underlying probabilistic framework. Furthermore, the training objective $\mathcal{L}_{\text{Vis}}$ is meticulously formulated to minimize the discrepancy between the true noise $\epsilon$ and the predicted noise $\epsilon_\theta$, thereby ensuring effective model training.

## C. Experiment Details

### C.1. Visual Adapter

We create a trainable copy of the pretrained UNet encoder and middle block, denoted as $U_{\text{copy}}$. While ControlNet employs an additional convolutional neural network to map control images (e.g., Canny edges or depth maps) from the pixel domain to the latent space, our method eliminates the need for such an extra network due to the prior handling of latent features. Additionally, following ControlNet-XS (Zavadski et al., 2024), we design the copied encoder and middle block in accordance with the ControlNet-XS Type B architecture, as illustrated in Fig. 18.

### C.2. Perceptual Fidelity Optimizer

The Pfo follows the completion of semantic residual retrieval. During optimization, we use the AdamW optimizer (Loshchilov & Hutter, 2017) with a learning rate set to 0.3. We balance the time cost and reconstruction quality by using 500 steps for optimization, which takes approximately 175s for a Kodak 768x512 image. During the optimization process, the time steps $n$ are not randomly selected from all 0 to 1000 as in the forward process of the diffusion model. Instead, they are chosen based on our denoising steps. For instance, if we use 4-step DDIM sampling for denoising, our optimization process specifically targets the noise levels at these 4 steps. For more visual results, see Figure 22. And, we perform inference every 50 optimization steps and select the result with the lowest LPIPS (Zhang et al., 2018) to ensure the fidelity.

*Table 6.* Detailed complexity of each component.

| | **Latent** | **Semantic Residual retrieval** | | **Diffusion** (4steps/3steps) | **Total** |
|---|---|---|---|---|---|
| | | Srr (+Pfo) | Index Coding | | |
| Enc (s) | 0.10 | 5.77 (+175.29) | 0.0015 | - | 5.87 (+175.29) |
| Dec (s) | 0.06 | - | 0.0004 | 0.54/0.37 | 0.60/0.43 |

## C.3. Complexity Evaluation

In Table 6, we provide the complexity results of each part. The Pfo module takes most of the encoding time for its iterative updating process. The total decoding speed remains competitive with existing methods. As shown in Table 7, complexity comparison with existing methods are listed. These data were all tested on the Kodak dataset using an RTX 4090 GPU.

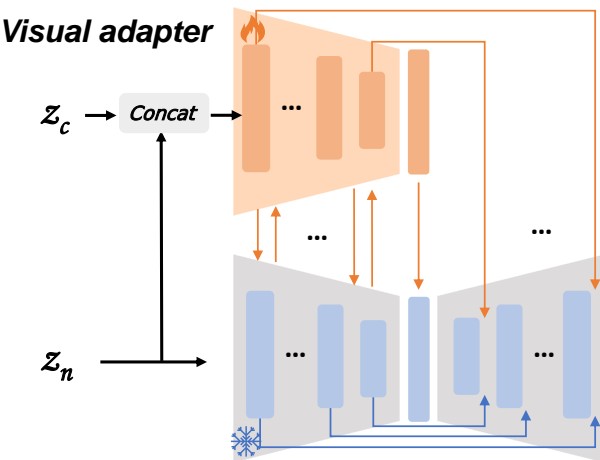

*Figure 18.* The architecture of Visual Adapter.

*Table 7.* Complexity Comparison of different methods based on denoising steps, encoding speed, decoding speed. 'w/o SRC' indicates removing both MLLM and Pfo. * indicates the extra time required for retrieving the full image caption.

| Type | Method | Denoising steps | Encoding Speed (s) | Decoding Speed (s) |
|---|---|---|---|---|
| VAE-based method | Cheng2020 | – | 2.86 | 6.69 |
| | ELIC | – | 0.057 | 0.079 |
| GAN-based method | MS-ILLM | – | 0.038 | 0.059 |
| | HiFiC | – | 0.036 | 0.061 |
| Diffusion-based method | DiffEIC | 50 / 20 | 0.128 | 4.57/1.96 |
| | PerCo | 20 / 5 | 0.08 (+ 0.32)* | 2.13/0.64 |
| | Text-Sketch | 25 | 62.045 | 12.028 |
| | Ours w/o SRC | 4 / 3 | 0.10 (+ 3.24)* | 0.60/0.43 |
| | Ours | 4 / 3 | 181.16 | 0.60/0.43 |

## C.4. Other Implementation Details

*1) Details of Baseline Model*: To ensure a fair comparison, we evaluated all methods as follows: For open-sourced approaches (e.g., DiffEIC, PerCo, CDC, MS-ILLM, Text-Sketch), we used their publicly available pretrained models. For non-open-sourced methods, we relied on the official results reported in their respective publications (e.g GLC, CorrDiff). Additionally, since the open-sourced HiFiC model operates at a higher bitrate, our data was sourced from the reproduced

version by the DiffEIC's author. For PerCo, the performance of the open-sourced version is slightly different with the offical results, so we provided comparison with both version.

*2) Training*: The Feature Compressor and the Visual Adapter models which is based on the stable diffusion v2.1 are trained on the LSDIR (Li et al., 2023) and Flicker2w (Liu et al., 2020) dataset. We first preprocess this dataset using LLaVA (Liu et al., 2024a) to obtain a caption corresponding to each image. Note that this training process is divided into two stages. In the first stage, we set $\lambda_d$ and $\lambda_p$ to 0, and $\lambda_R$ to $\{24, 14, 4, 2, 1\}$, training for 150K iterations. In the second stage, we set $\lambda_d$ to 1, $\lambda_p$ to $\{0.4, 0.6, 0.8, 1, 1\}$, and $\lambda_R$ to $\{36, 16, 6, 3, 1.5\}$, training for 100K iterations. During training, we center-crop images to a dimension of $512 \times 512$ and randomly set 30% of the captions to empty strings to enhance the model's generative capabilities and sensitivity to prompts.

As shown in Eq. (18), the term $\mathcal{L}_R = R(z_c)$ is estimated by a probability estimation model with $p_{\hat{y}}$ during training, which is formulated as $R = \mathbb{E}[-\log_2(p_{\hat{y}})]$. Here, $\hat{y}$ represents the output obtained by passing $z_0$ through the feature encoder followed by quantization.

*3) Semantic Residual Retrieval.* After the training of the Latent Compressor is finished. The decoded image $x'$ and its corresponding original image $x$ are used to extract the semantic residual.

The prompt for GPT4o to extract the raw captions is:

- ***"Please describe this picture in detail with 40 words. Do not provide any description about feelings."***

Then we use the captured $\mathbf{f}_{mllm}(x)$ and $\mathbf{f}_{mllm}(x')$ to further capture the residual information $c_{\mathbf{res}}$. The prompt used for GPT-4o is:

- ***"Original Image: 'f$_{mllm}(x)$'; Compressed Image: 'f$_{mllm}(x')$'. Provide information that is in the original image but not included in or mismatch with the compressed image. Don't include information that is already in the compressed image. Please use most compact words. Do not include the description for the compressed image. For example: if input is Original Image: A red barn surrounded by trees, reflected in a pond. Compressed Image: red house surrounded by trees. Residual caption is : A barn reflected in a pond. Please refer to this to output. Do not appear words like 'compressed image', 'original image' and 'The semantic residual is'. If you think that the two descriptions mean almost the same thing, please output an empty string. "***

## D. More Results

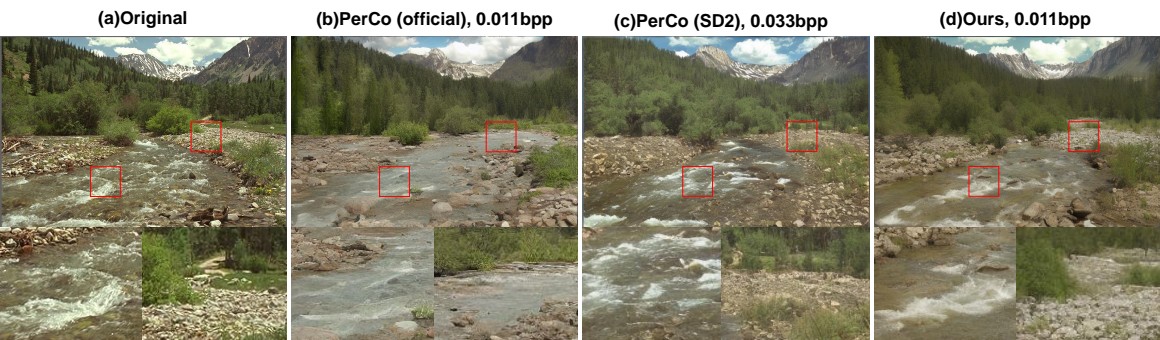

*Figure 19.* Visual comparison with the official PerCo version.

In Figure 19, the reproduced version of PerCO (marked as PerCo(SD2)) provides better visual quality than the original one. However, in terms of objective metrics in Table 8, the official PerCo version performs better. Therefore, we provide comparisons with both versions to demonstrate our method's efficiency.

In Figure 23, we also provide more examples of semantic residual by MLLM. In Figure **??**, While each metric improves with its respective optimization, PSNR-optimized results tend to produce smoother textures, such as in water and trees, whereas LPIPS and MS-SSIM optimizations are better at preserving details.

*Table 8.* Comparison with results in PerCo's original paper in terms of BD-Rate. Note that the PerCo (official) is also LPIPS optimized. Our method still shows significantly better performance.

| Methods | BD-Rate (%) / Kodak | | |
|---|---|---|---|
| | **LPIPS** | **PSNR** | **MS-SSIM** |
| PerCo(official) | 0 | 0 | 0 |
| PerCo(SD2) | 21.7 | 88.3 | 15.4 |
| Ours w/o Pfo | -34.5 | -54.7 | -33.4 |
| Ours | -41.5 | -52.0 | -32.7 |

In Figure 20, 21, and 24, we present visualizations for different bpp values. It can be observed that our method achieves significantly impressive subjective quality.

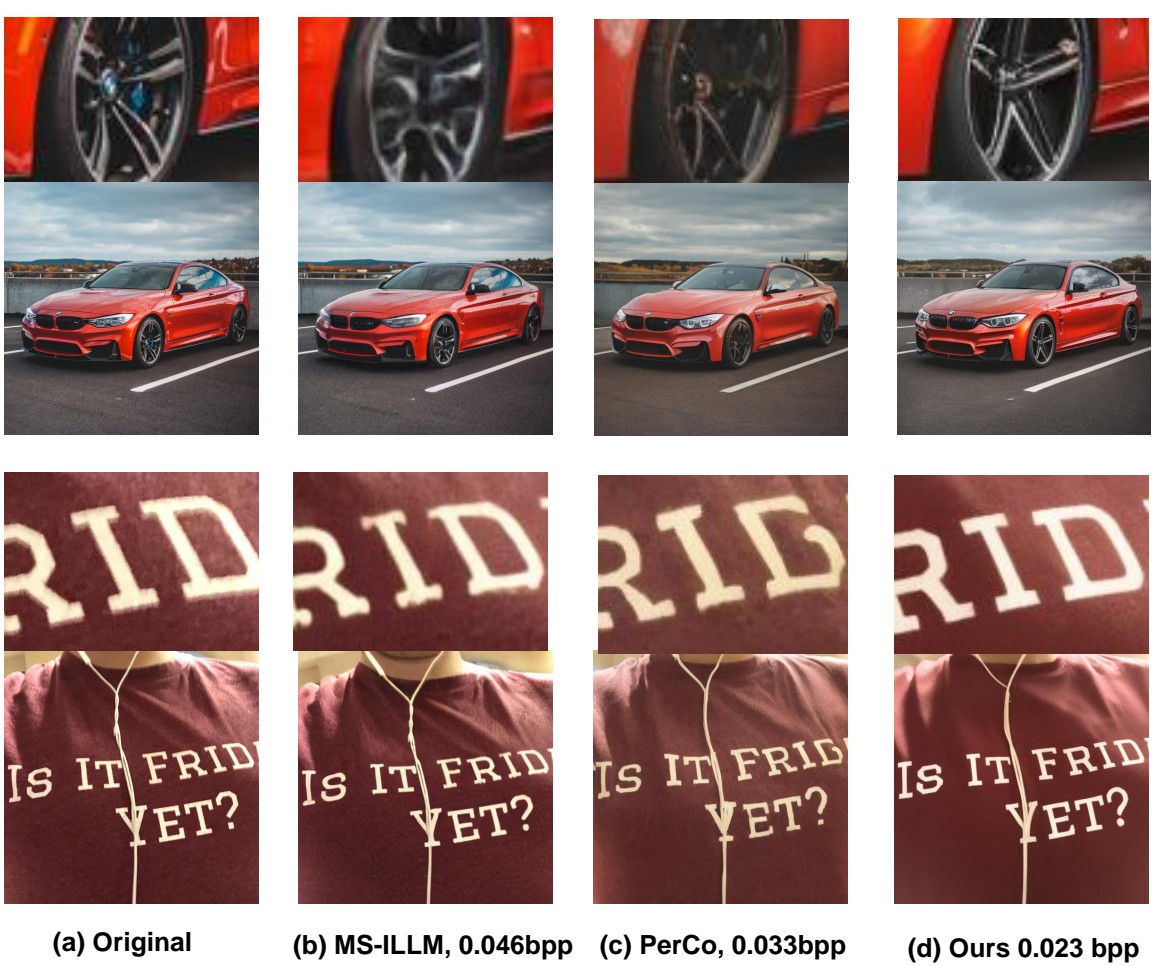

(a) Original      (b) MS-ILLM, 0.046bpp    (c) PerCo, 0.033bpp     (d) Ours 0.023 bpp

*Figure 20.* Visual comparison at low bpp.

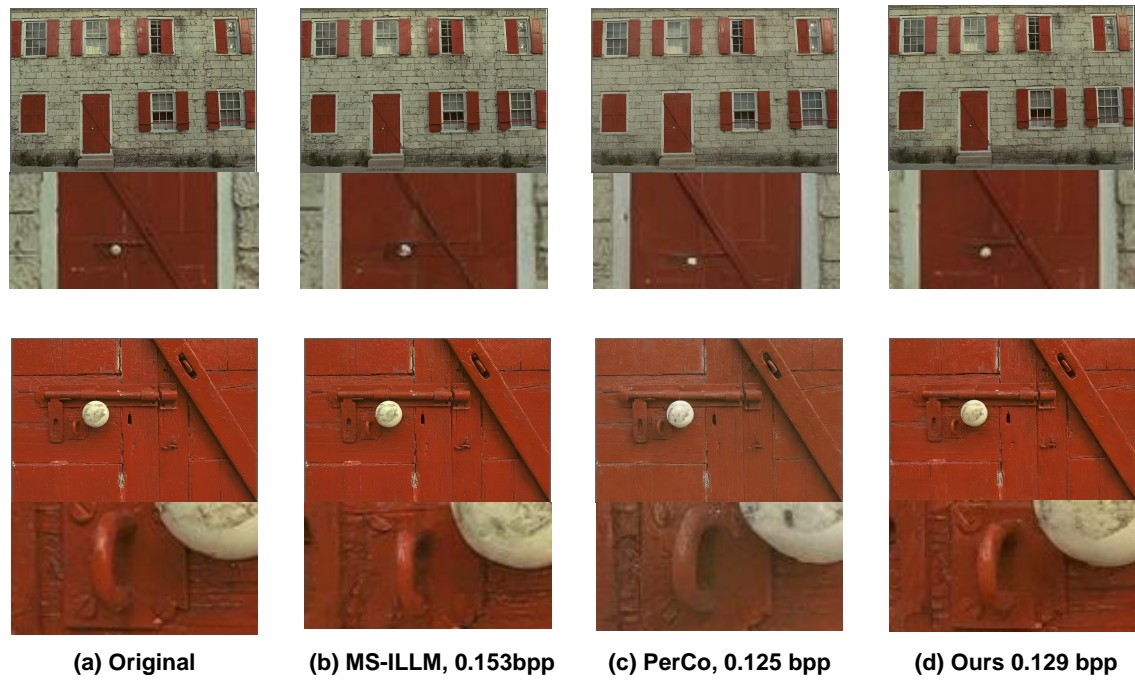

*Figure 21.* Visual comparison at high bpp.

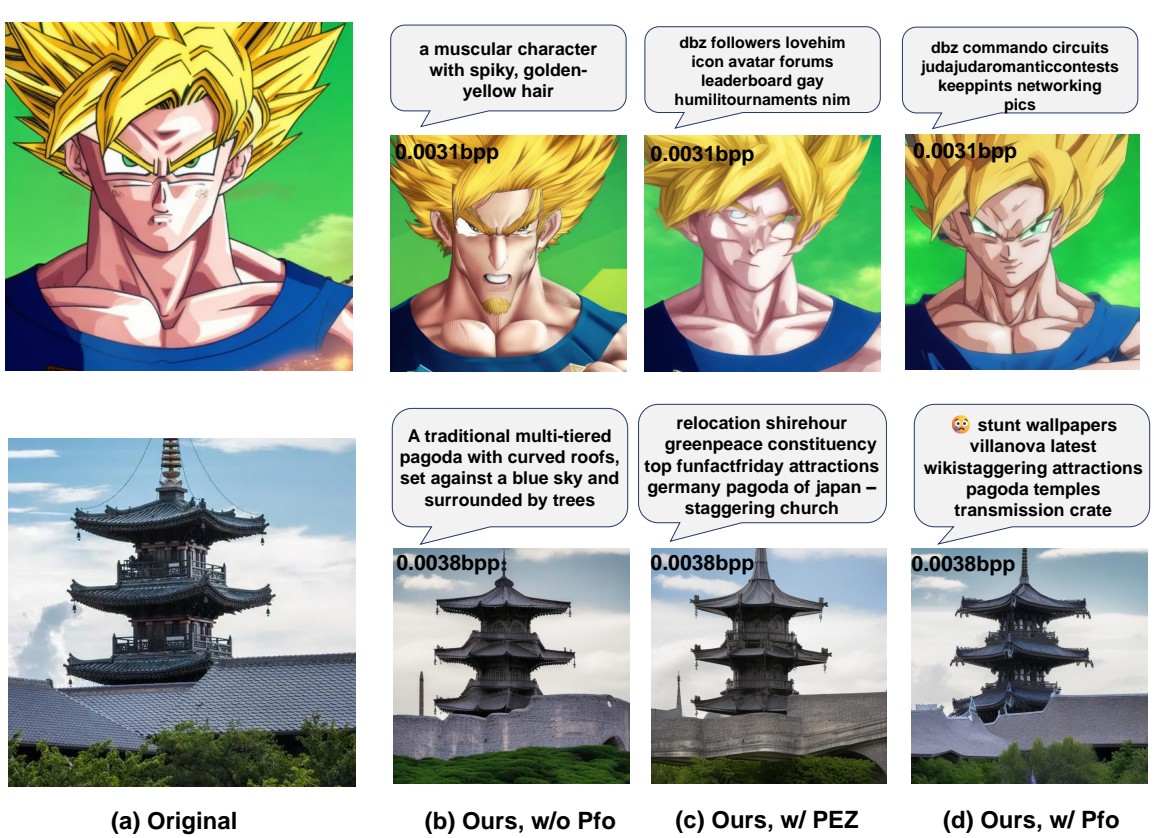

*Figure 22.* Effectiveness of Pfo optimization and corresponding texts

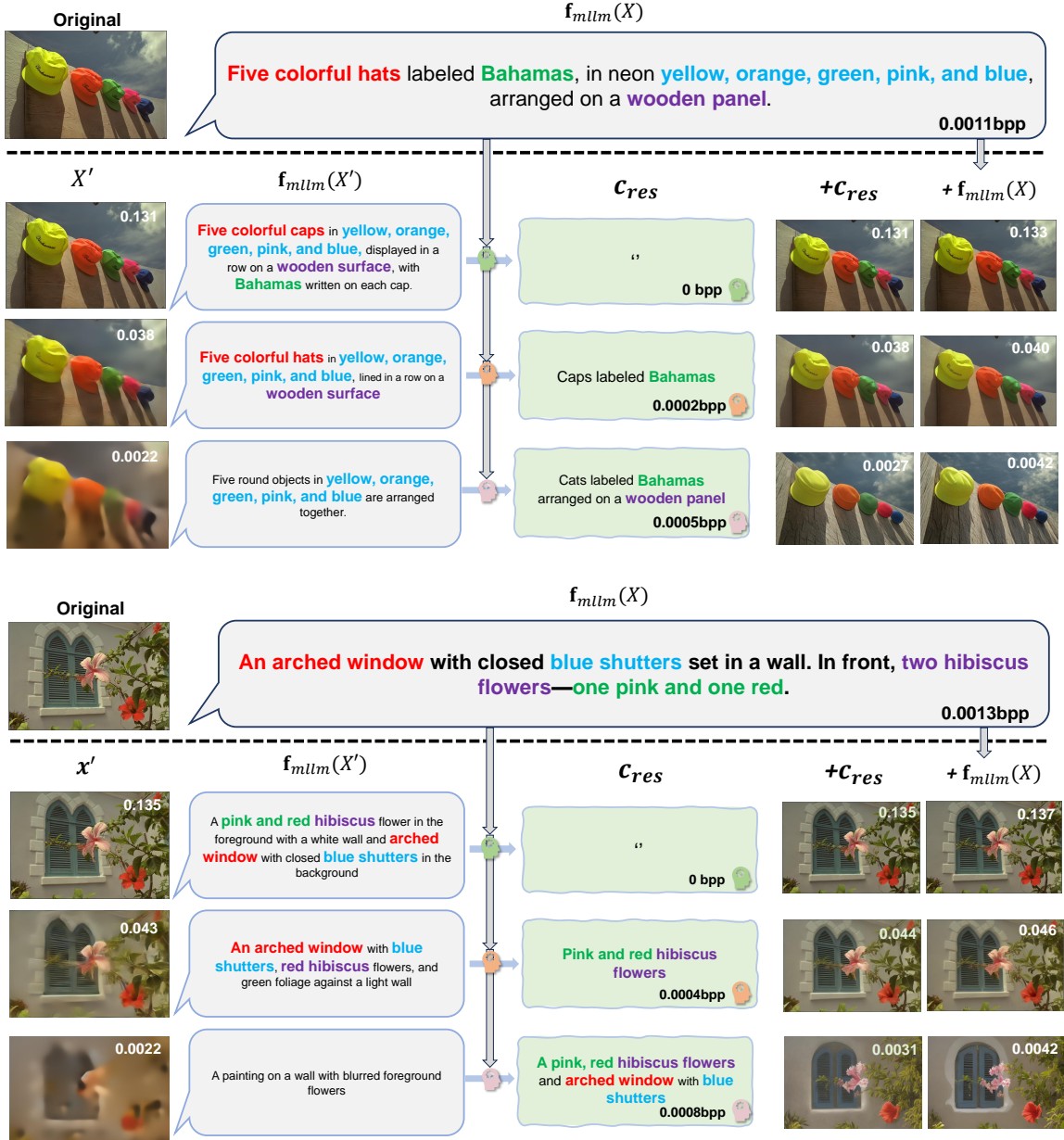

*Figure 23.* More visualization examples of MLLM-based Semantic Residual Retrieval.

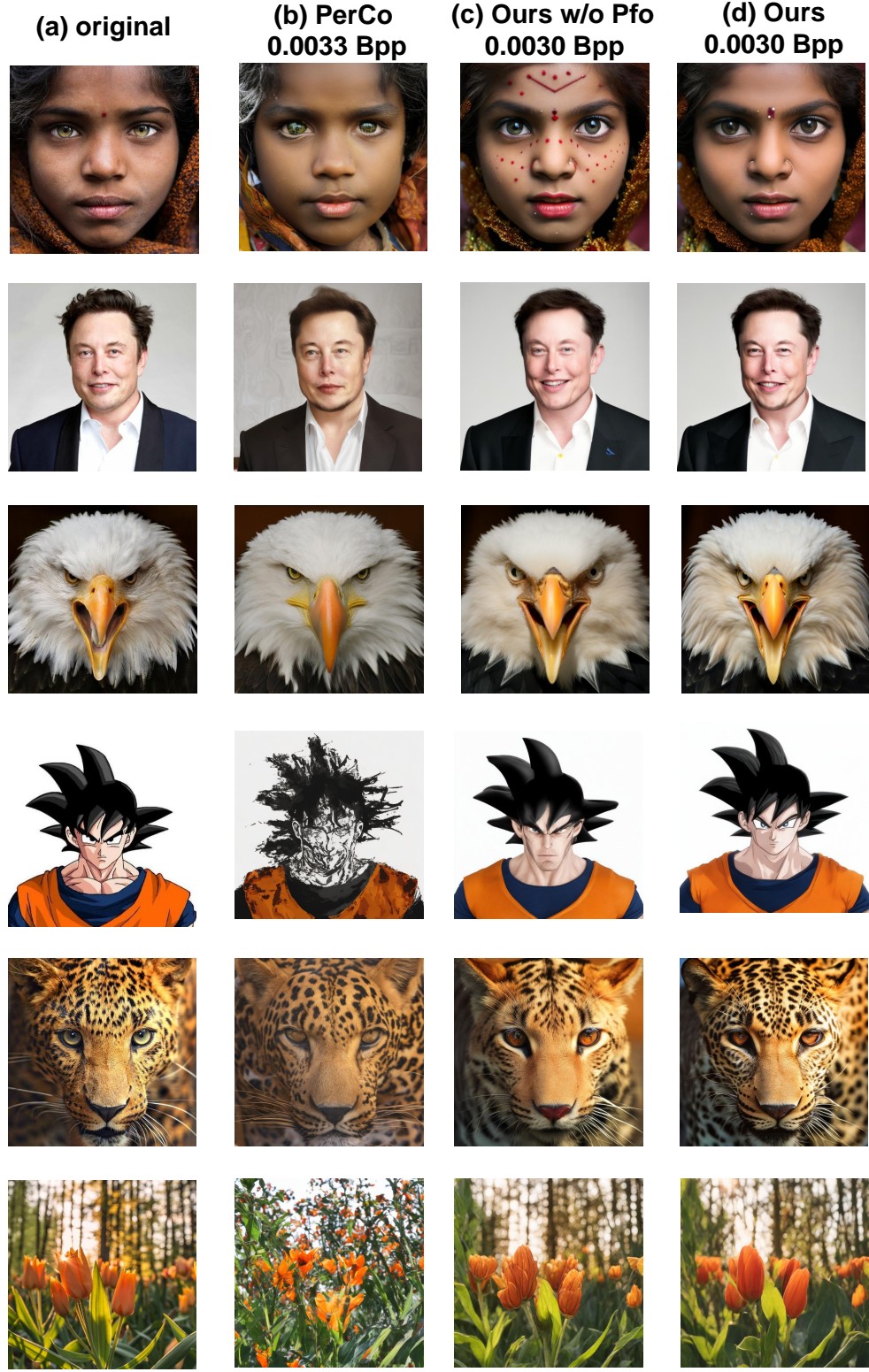

*Figure 24.* More subjective visual comparisons

