# OpenReview forum: "Ultra Lowrate Image Compression with Semantic Residual Coding and Compression-aware Diffusion"
_ICML.cc/2025/Conference — ICML 2025 poster_

### Official Review · Reviewer_PLiJ · 2025-03-10

**Overall Recommendation:** 3

**Summary:**

The paper introduces an ultra-low rate image compression method combining Semantic Residual Coding and a Compression-aware Diffusion Model. SRC efficiently encodes semantic differences between original and compressed latent images, minimizing redundancy, while CDM aligns diffusion steps with compression levels to improve reconstruction fidelity. Experiments demonstrate that ResULIC surpasses existing diffusion and GAN-based methods, achieving up to 80.7% and 66.3% BD-rate savings in LPIPS and FID metrics, respectively, particularly excelling at ultra-low bitrates.

**Claims And Evidence:**

Most claims are supported by experimental evidence regarding Semantic Residual Coding and the Compression-aware Diffusion Model, both validated by quantitative and qualitative analyses. However, the Pfo module increases encoding time (~175s/image), raising practical concerns not fully addressed.

**Essential References Not Discussed:**

N/A

**Experimental Designs Or Analyses:**

Yes, the experimental design and analyses are rigorous, with metrics and ablations. Minor issues include reliance on unofficial baseline implementations (e.g., PerCo) and insufficient clarity on differences from originals.

**Methods And Evaluation Criteria:**

The proposed methods and evaluation criteria are appropriately designed for ultra-low bitrate image compression. However, the practicality of the computationally expensive Perceptual Fidelity Optimization module (~175s/image) warrants further clarification.

**Other Comments Or Suggestions:**

N/A

**Other Strengths And Weaknesses:**

Strengths:
- combination of existing ideas
- clearly presented and significant

Weaknesses:
- high computational cost

**Questions For Authors:**

- Given the high computational cost of PFO, can you clarify practical use cases?
- Could differences between your reproduction and the official PerCo implementation affect your comparisons?

**Relation To Broader Scientific Literature:**

The paper combines diffusion-based generative models (PerCo, DiffEIC), multimodal semantic retrieval (MLLMs), and residual coding into a unified compression framework.

**Theoretical Claims:**

The theoretical claims and proofs look correct and are clearly presented. However, I did not look into the details very much.

---

> ### Author Rebuttal · Authors · 2025-04-01
>
> Thank you for your kind recognition and these insightful questions—they are indeed crucial for understanding our work.
>
> Q1: **Regarding the computational cost of PFO and practical use cases**
>
> A1: For PFO, our goal is to demonstrate the potential of prompt optimization in further improving the overall fidelity, even though it is currently computationally intensive.
> Additionally, **the PFO introduces no additional latency during decoding**, making it acceptable for performance-critical scenarios, such as cloud-based restoration services where computational resources are readily available and the encoding latency is not strictly constrained.
> However, even without PFO, our method still achieves superior performance compared to existing approaches (as demonstrated in Table 1). This underscores the effectiveness of our core framework. Furthermore, we are actively working on optimizing PFO's efficiency to facilitate its wider practical deployment in future applications.
>
> Q2: **Reproduction vs. official PerCo implementation**
>
> A2: Since the official PerCo code is not publicly available, we mainly compared with a publicly avaiable version. This reproduced version provides some drop in metrics like LPIPS/MS-SSIM, but better at FID/KID and visual quality.
> Notbaly, to ensure a fair comparison with both versions: We have also benchmarked against PerCo’s official  objective results (in Table 9) and subjective comparisons (in Fig. 20) in Appendix D. We also list the table below. Our method also consistently outperforms PerCo (offical), even when comparing with its strongest metrics like PSNR/LPIPS/MS-SSIM.
>
> |||BD-Rate(%)↓|||
> |:-:|:-:|:-:|:-:|:-:|
> |Method|LPIPS|PSNR|MS-SSIM|
> |PerCo(offical)|0|0|0|
> |PerCo(SD2.1)|21.7|88.3|15.4|
> |**ResULIC W/o Pfo**|-34.5|-54.7|-33.4|
> |**ResULIC**|-41.5|-52.0|-32.7|

---

### Official Review · Reviewer_5kJu · 2025-03-11

**Overall Recommendation:** 4

**Summary:**

This paper introduces ResULIC, a novel framework for ultra-low-bitrate image compression that integrates semantic residual coding (SRC) and a compression-aware diffusion model (CDM). SRC is proposed to capture the semantic disparity between the original image and its compressed latent representation. CDM is used to establish a relationship between diffusion time steps and compression ratios, improving reconstruction fidelity and efficiency. Compared to the state-of-the-art diffusion-based compression method PerCo, ResULIC achieves superior rate-distortion performance.

**Claims And Evidence:**

Yes.

**Essential References Not Discussed:**

No

**Experimental Designs Or Analyses:**

Limited real-world testing or user studies. While objective metrics like LPIPS and FID are useful, human perceptual studies (e.g., Mean Opinion Score) would provide additional validation of realism. This is not a major issue, but including user evaluations would further strengthen the claims.

**Methods And Evaluation Criteria:**

Yes.

**Other Comments Or Suggestions:**

No.

**Other Strengths And Weaknesses:**

Strengths:
1. The idea of using MLLM-based retrieval to encode only missing semantic details rather than full textual descriptions is an innovative and practical refinement.

2. CDM dynamically adjusts diffusion steps based on compression levels, improving efficiency without needing extensive modifications to pre-trained models.

3. Strong experimental results and relevance to ultra-low bitrate applications.

4. Comprehensive experiments and ablation studies.


Weaknesses:

1. Why was Stable Diffusion v2.1 specifically chosen as the backbone? If the latest SOTA diffusion model FLUX is used, is possible to further improve the compression performance.

2. Limited real-world user studies: While ResULIC shows strong performance in perceptual metrics (FID, LPIPS), there are no human perceptual studies. Real-world testing on actual user preference rankings could validate if the perceived quality aligns with objective metrics.

**Questions For Authors:**

No.

**Relation To Broader Scientific Literature:**

1. ResULIC builds directly on recent research in learned image compression (Balle et al. (2017, 2018), HiFiC (Mentzer et al., 2020)) , multimodal generative models (PerCo (Careil et al., 2024)), and diffusion-based restoration (ResShift (Yue et al., 2024), RDDM (Liu et al., 2024)).
2. It differentiates itself by combining residual-based multimodal compression with diffusion-aware bitrate modeling, which has not been explicitly addressed before.
3. Its evaluation approach is aligned with state-of-the-art perceptual fidelity methods, reinforcing its relevance in the field.

**Theoretical Claims:**

Yes.

---

> ### Author Rebuttal · Authors · 2025-04-01
>
> We sincerely appreciate the reviewer’s insightful comments and recognition of our work’s potential. The questions raised are highly valuable for improving the robustness and impact of our research.
>
> Q1: **Implementation details about compared methods**
>
> A1: Thank you for your reminder. To ensure a fair comparison, we evaluated all methods as follows:
> For open-sourced approaches (e.g., DiffEIC, PerCo, CDC, MS-ILLM, Text-Sketch), we used their publicly available pretrained models. For non-open-sourced methods, we relied on the official results reported in their respective publications (e.g GLC, CorrDiff).
> Additionally, since the open-sourced HiFiC model operates at a higher bitrate, our data was sourced from the reproduced version by the DiffEIC's author. For PerCo, the performance of the open-sourced version is slightly different with the offical results, so we provided comparison with both version.
>
> Q2: **Choice of Stable Diffusion v2.1 as Backbone**
>
> A2: Thank you for raising this insightful question. It is possible and in line with expectations that  larger and better diffusion models can have improved performance. The choice of Stable Diffusion v2.1 as the backbone was primarily driven by two key considerations:
> - Fair Comparison with Existing Works: Current state-of-the-art methods (e.g., PerCo, DiffEIC) adopt SD v2.1 as their base model. To ensure a direct and equitable evaluation of our method’s performance improvements, we maintained consistency in the backbone architecture.
> - Complexity-Performance Balance: While advanced models like FLUX could potentially enhance compression performance, their large parameter size (12B) and high GPU memory usage make them impractical for real-world applications for image compression. Therefore, we opted to use smaller models to validate the reliability of our method.
>
> We fully acknowledge that leveraging newer diffusion models is a promising direction for further improvements. In fact, we are actively working to secure additional computational resources to explore such extensions in future research. At the same time, we are also investigating how to better utilize the quantized version of FLUX to enhance our performance. Your suggestion aligns perfectly with our roadmap for advancing this work.
>
> Q3: **User Studies and Perceptual Validation**
>
> A3: To fully demonstrate the comparison of subjective quality, we have included some example images in the main text and appendix. However, we fully acknowledge that user studies are indispensable for comprehensive validation.
>
> After concentrated testing efforts, we're pleased to share some initial subjective user study results(https://anonymous.4open.science/api/repo/public-E45F/file/user_study.png?v=0cb25072). Due to time constraints, we conducted a preliminary evaluation at two bitrate points with 20 participants. Each participant assessed randomly selected images from the DIV2K test set (30 images per bitrate point). The final results were calculated as the average of all votes. Subjective tests confirm our method's clear visual advantage over baselines. We will keep working on providng a more comprehensive user studies.
>
> Thank you again for the constructive feedback—we will incorporate these suggestions to strengthen our research further.

---

### Official Review · Reviewer_XQ42 · 2025-03-12

**Overall Recommendation:** 3

**Summary:**

The present paper describes a perception-oriented image compression framework that utilizes the residual multi-modal semantics (w.r.t what’s could be recovered by the latent decoder) as guidance conditioning for the diffusion denoising process towards improved fidelity at lower rate costs. The insight that fidelity improves with increased number of denoising steps, for a given bitrate, the paper also describe a variable compression level method with improved flexibility. The method differs from prior MLLM-based image compression methods by accounting for the semantic redundancy between the image latent representations and multi-modal embeddings, thus yielding improved rate-perception trade-off.

**Claims And Evidence:**

The empirical evidence and ablative analysis are comprehensive and could adequately support the claims. It makes sense to steer the denoising diffusion process with only semantic residuals and the experimental results look promising.

**Essential References Not Discussed:**

No essential references missing to the best of my knowledge.

**Experimental Designs Or Analyses:**

I have checked the soundness of experimental designs and analyses in the main text as well as the supplementary materials. Please refer to  **Methods and Evaluation Criteria** section for my questions.

**Methods And Evaluation Criteria:**

The proposed methods and evaluation criteria, for the most part, make sense for the problem at hand. The only issue is, I wonder what is the difference between conditioning the diffusion model directly with visual cues instead of explicit texts? The semantic residual is expected to be of high-frequencies whose encapsulated information may not always be `conveyable' via textual descriptions. Would there be information loss during semantic residual retrieval? Also, it could be seen from Figure 5 in the main text and, for instance, Figure 23 in the supplementary materials, that the textual prompts could contain gibberish that could not be comprehended by human. Why would such prompts be beneficial to guiding the denoising process?

**Other Comments Or Suggestions:**

= I think Figure 5 is not properly referred to in the main text.

**Other Strengths And Weaknesses:**

N/A.

**Questions For Authors:**

Please refer to  **Methods and Evaluation Criteria** section for my questions.

**Relation To Broader Scientific Literature:**

They are related to foundational model-based generation, in particular conditional diffusion models and multi-modal vision language models.

**Theoretical Claims:**

I have checked the correctness of the proofs. They look sensible to me.

---

> ### Author Rebuttal · Authors · 2025-04-01
>
> Q1: **What is the difference between conditioning the diffusion model directly with visual cues instead of explicit texts**
>
> A1: In our method, explicit texts primarily compensate for semantic gaps in the visual condition, especially at ultra-low bitrates. For example:
> - **Low-bitrate regime**: In extreme low-bitrate scenarios where compressed features may retain only basic structural information (e.g., object locations and edges), relying solely on these limited visual cues can lead to vital semantic mismatch (e.g cat to dog).  Our method adds text descriptions to fill in missing details, bridging the semantic gap from heavy compression. The subjective results can be viewed at: https://anonymous.4open.science/api/repo/public-2D3D/file/test2.jpg?v=1ef61402
> - **High-bitrate regime**: As the bitrate increases, the visual condition retains sufficient information, and the reliance on text gradually diminishes. This is empirically validated in Figures 9,10 of our paper, where the residual text length drops to zero at higher bitrates.
> Thus, visual conditions and texts are synergistic: the former anchors structural fidelity, while the latter contains compact high-level semantics (visual cues contain richer and preciser information, while texts are more compact). Our hybrid solution can utilize both sources to achieve overall improved performance.
>
> |w/o text||w/ text||
> |:-:|:-:|:-:|:-:|
> |Bpp|LPIPS(↓)|Bpp|LPIPS(↓)|
> |0.002163|**0.496696**|0.002681|**0.468738**|
> |0.004901|**0.403217**|0.005481|**0.386965**|
> |0.021926|**0.205847**|0.022849|**0.232723**|
> |0.062561|**0.135126**|0.063262|**0.123806**|
>
> Q2: **Would there be information loss during semantic residual retrieval**
>
> A2: During the process of semantic residual retrieval, what we extract is not directly the semantic content of residual features, but rather, as illustrated in Figure 6, we use MLLM to extract the residual information between the original image $x$'s caption and caption of the compressed image $x'$: $caption_{res}=MLLM(x)-MLLM(x')$, instead of $MLLM(x-x')$ . The table below shows the results of semantic residual retrieval for different models (GPT-4o, Llama-3.2-11B) in Kodak.
>
> Therefore, the semantic residual in the form the text is usually conveyable as long as the text description matches with the input image. In terms of text-level information loss, it mainly depends on the capability of the underlying MLLM models as shown here in the table.
>
> |w/o Srr||W/ Srr(GPT-4o)||W/ Srr(Llama-3.2-11B)||
> |:-:|:-:|:-:|:-:|:-:|:-:|
> |Bpp|LPIPS(↓)|Bpp|LPIPS(↓)|Bpp|LPIPS(↓)|
> |0.005189|**0.476899**|0.00372|**0.468738**|0.003737|**0.493859**|
> |0.008308|**0.386744**|0.006777|**0.386965**|0.006969|**0.390045**|
> |0.033089|**0.233531**|0.031589|**0.232723**|0.031614|**0.233985**|
>
> From the table, we can observe that since GPT-4o possesses stronger capabilities, the LPIPS between reconstructions after SRR and those from the original captions remains nearly identical. In contrast, Llama-3.2-11B shows some degradation, suggesting potential loss of certain keywords in the process.
>
> However, as you have rightly considered, the semantic residual represented in texts may not perfectly align with the pixel-level residual signal. Hence, we proposed the PFO module to compensate for the feature information that natural language cannot describe. Please correct us if we misunderstand the question you raised.
>
> Q3: **Why would such “gibberish” prompts be beneficial to guiding the denoising process**
>
> A3: Thank you for raising this insightful question. Regarding this issue, we provide the following clarification of the PFO pipeline:
> A word is first converted into an embedded token space, and the PFO performs search in the token space through gradient backpropagation guided by output image reconstruction quality. The token vector is iterally updated and finally converted back as a new word. Semantic control in diffusion models operates through CLIP's text embedding space rather than surface-level language.
>
> PFO performs gradient-based search in this latent token space, optimized solely for image reconstruction quality without explicit linguistic constraints. Therefore, the PFO may break the grammar and readability of the word. Consequently, the generated captions may not always exhibit high readability (though still retaining interpretable tokens such as "DBZ (Dragon Ball Z)," "pagoda," and "temples," as shown in Figure 23).
>
> However, they can achieve higher semantic alignment with the original image in CLIP's embedding space. The high-dimension token space encapsulates information beyond the expressive capacity of human-readable language, thereby enabling more effective guidance.
>
> Q4: **Figure 5 is not properly referred to in the main text**
>
> A4: Thank you for your reminder. The principle of Figure 5 is indeed consistent with our response to **Q3**, and we will subsequently add a detailed explanation in the main text.

---

### Official Review · Reviewer_jfbj · 2025-03-14

**Overall Recommendation:** 2

**Summary:**

This paper proposes a diffusion model-based image compression method. First, a pretrained codec is used to obtain a latent representation, which guides the generation process of the diffusion model. Second, semantic information is introduced as additional guidance. To reduce the overhead of transmitting semantic information, only the residual between the ground truth and the decoded image’s semantics is transmitted. Experimental results show that on CLIC20, the proposed method achieves a lower FID compared to other approaches. For this paper, I cannot provide an evaluation of the novelty of the diffusion model part, as I am not very familiar with diffusion models. My main concern lies in the model's complexity and its comparison with GLC.

**Claims And Evidence:**

Yes

**Essential References Not Discussed:**

I believe this paper provides a sufficient discussion of related work but lacks coverage of distortion-oriented image compression methods.

**Experimental Designs Or Analyses:**

Yes.

**Methods And Evaluation Criteria:**

As a perceptual quality-optimized image compression method, it is reasonable for the authors to use FID, LPIPS, and DISTS as evaluation metrics.

**Other Comments Or Suggestions:**

line 1033:  which takes approximately **175s minutes** for a Kodak

**Other Strengths And Weaknesses:**

This paper achieves strong performance in terms of FID. Compared to other diffusion-based methods, it requires fewer iteration steps, resulting in a faster decoding speed. However, I believe its decoding speed is still slower than GAN-based methods and comes with higher computational complexity. According to Figure 15, its LPIPS performance is not as good as the GAN-based GLC, yet it comes with higher complexity.

For evaluation on the CLIC2020, DIV2K, and Tecnick datasets, the authors followed the approach of CDC (Yang & Mandt, 2022) by resizing images to a short side of 768 and then center-cropping them to 768×768. I wonder if the authors could provide results for full-resolution inference, as I am curious about the memory overhead of the diffusion model during full-resolution inference.

In the GLC paper, the FID evaluation method is consistent with that of HiFiC. The authors should compare their method with GLC.

I am generally leaning towards borderline.

**Questions For Authors:**

How to understand the result in Table 4 where fewer steps lead to better performance?

**Relation To Broader Scientific Literature:**

Researchers working on integrating diffusion models into image compression may find this paper interesting.

**Theoretical Claims:**

I am not very familiar with diffusion models, but I reviewed the proof as thoroughly as possible.

---

> ### Author Rebuttal · Authors · 2025-04-01
>
> Thank you for your thoughtful feedback. Regarding your inquiry, we would like to provide the following clarification:
>
> Q1: **Comparing with GAN-based GLC**
>
> A1: We highlight our advantage over GLC as below:
> 1. **Ultra-low Rate Support**: GLC is a very competitive and representative GAN-based compression. However, similar to other GAN-based approaches (e.g., MS-ILLM, HiFiC), it faces challenges at ultra-low bitrates. Although GLC achieved bitrates below 0.01 bpp on datasets like CelebAHQ, such performance requires task-specific retraining for facial images, limiting its generalizability. For general natura images (e.g. Kodak, CLIC), GLC's minimum achievable bitrate is approximately 0.02 bpp, whereas our method remains effective even at an ultra-low bitrate of 0.0026 bpp. **As the bitrate decreases further, the effectiveness of feature-only optimization gradually diminishes.**
> 2. **Simplified and Stable Optimization**: GLC employs GAN and a complex 3-stage training for refining compressed features. In contrast, our approach achieves superior visual quality and competitive objective results currently with simple ***MSE-optimized latent compressor***, and mainly focusing on how diffusion models can effectively enhance these features. From the perspective of feature compression, GAN based GLC, diffusion based PerCO, and our method share similar principle—using a pretrained VQGAN/VAE to perform image compression the latent features. However, the optimization strategies diverge.
>
> In essence, GAN-based and Diffusion-based compression frameworks are two competitve pathes. These two approaches are highly complementary and can mutually reinforce each other.  Combining GLC's feature refinement with diffusion recovery could further advance ultra low-bitrate image compression performance.
>
> Q2: **Full resolution Memory Cosumption**
>
> A2: We fully understand this concern and would like to provide the following clarifications:
> 1. **Test Setting**: As mentioned by the reviewer, our current test setting is to align with existing diffusion-based methods such as CDC and DiffEIC for fair comparison.
> 2. **Memory Benchmark**: On full-resolution CLIC images (2048×1365):
>   - while GLC's closed-source nature prevented direct evaluation, its backbone VQGAN's peak memory usage is 11.2GB (w/o compression module of GLC)
>   - Our method's peak usage is 17.8GB (at the Diffusion's VAE decoder).
>
> This marginal increase in memory usage represents a reasonable tradeoff. Considering our method's unique capability of achieving ultra-low bitrate compression and the potential of diffusion-based frameworks, we believe they hold promise for substantial advancements in both performance and efficiency.
>
> Q3: **Full resolution results and Comparing with GLC about FID**
>
> A3: As mentioned, our method demonstrates more promising performance at ultra low rates (<0.01bpp). GLC’s paper evaluates only a narrow bitrate range (0.02-0.03bpp), we matched this range for fair comparison on CLIC and DIV2K (full resolution) and MS-COCO 30K (256×256, following GLC’s setup). The results can be seen at https://anonymous.4open.science/api/repo/public-5C7B/file/comparison.png?v=f3d457fd. As can be seen from the figure, our method still provides better or competitive results over GLC for full resolution tests.
>
> Q4: **Question about Table 4: Fewer steps but better performance**
>
> A4: In Table 4, we provided two baselines for comparison. One is a strategy ANS used in previous works, the other is our ResULIC without the CDM strategy. The results showed that with **CDM** and our **Predicted-$x_0$ training strategy**, better performance can be achieved with fewer steps.
>
> More explanation of these two modules:
>
> **CDM**: CDM accelerates denoising by initiating diffusion from a distribution conditioned on compressed feature means $z_c$ rather than pure noise (Eq.8). This intermediate starting point reduces sampling steps while enhancing fidelity.
>
> **Predicted-$x_0$ Strategy**: Our sampling process (Algorithm 2 in the Appendix) suffers from error accumulation when deriving $z_0$ from noise predictions. The training strategy predicts $x_0$ to reduce error accumulation from noise prediction, improving accuracy and enabling high-fidelity reconstruction with fewer steps.
>
> Additional experiments are provided in the table to demonstrate reduced decoding delay and improved performance（In "Denoising Step", the numbers represents sampling steps (50，20，4) for <0.05bpp and  (20，10，3) for ≥0.05bpp）:
>
> ||||||BD-Rate(%)↓|||
> |:-:|:-:|:-:|:-:|:-:|:-:|:-:|:-:|
> |Method|w/predicted-x_0 training|w/CDM|Denoising Steps|LPIPS|DISTS|PSNR|MS-SSIM|
> |ANS|×|×|50/20|0|0|0|0|
> ||×|×|50/20|4.6|5.3|6.8|47.1|
> ||×|√|20/10|-29.3|-35.4|-19.5|-10.6|
> |**ResULIC**|√|×|20/10|-25.8|-29.5|-25.3|-16.7|
> ||√|√|20/10|-58.4|-66.8|-48.7|-22.9|
> ||√|√|4/3|-54.4|-65.8|-49.7|-23.9|
>
> We appreciate your insights and hope this clarifies the trade-offs and unique contributions of our work.

---

### Decision · Program_Chairs · 2025-05-01

**Decision:**

Accept (poster)

**Comment:**

This paper receives one weak reject, two weak accept and one accept. The authors address most of the reviewers' concerns. I encourage the authors to revise the paper based on the comments from the reviewers.